# MotionGPT3: Human Motion as a Second Modality

**Bingfan Zhu**[1] **Biao Jiang**[2] **Sunyi Wang**[1] **Shixiang Tang**[4] **Tao Chen**[2]
**Linjie Luo**[3] **Youyi Zheng**[1†] **Xin Chen**[3†]
[1]Zhejiang University   [2]Fudan University   [3]ByteDance
[4]The Chinese University of HongKong
`https://github.com/OpenMotionLab/MotionGPT3`

## Abstract

With the rapid progress of large language models (LLMs), multimodal frameworks that unify understanding and generation have become promising, yet they face increasing complexity as the number of modalities and tasks grows. We observe that motion quantization introduces approximation errors that cap motion quality, while unifying discrete text and continuous motion within a single-stream backbone amplifies cross-modal interference. Motivated by recent multi-branch designs that separate signals from different modalities, we propose MotionGPT3, a bimodal motion–language model for both understanding and generation. MotionGPT3 encodes raw motion into a continuous latent space, thereby avoiding quantization-induced artifacts, while leveraging the semantic prior of pretrained language models. A dual-stream Transformer with shared attention preserves modality-specific routes while enabling controlled, bidirectional information flow, which reduces interference, stabilizing optimization, and empirically accelerates convergence without degrading fidelity. For multimodal joint training, a generate-then-align three-stage schedule further improves stability and limits cross-task interference. Experiments show that MotionGPT3 achieves 2× faster convergence in training loss and up to 4× faster convergence in validation, while maintaining state-of-the-art performance on standard motion understanding and motion generation benchmarks.

## 1 Introduction

Multimodal large language models (MLLMs) have recently achieved rapid progress in understanding and generation across text, images (Team, 2024; Wu et al., 2024a; Zhou et al., 2024), audio (Agostinelli et al., 2023; Copet et al., 2023; Liu et al., 2024), and video (Kondratyuk et al., 2023; Zhang et al., 2023a; 2024d). Built on the strong semantic priors and in-context learning capabilities of pretrained LLMs, these models capture long-range dependencies and compositional structure, enabling few-shot transfer and controllable across modalities (Alayrac et al., 2022; Chowdhery et al., 2023; Dong et al., 2023; Li et al., 2023; Touvron et al., 2023). Toward Unified Motion–Language Modeling. While most prior work has focused mainly on text-driven motion synthesis (Shafir et al., 2023; Tevet et al., 2022a;b;c; Xin et al., 2023; Zhang et al., 2024a), unified motion–language models for both understanding and generation remain comparatively underexplored. Pursuing both tasks in a single model demands representations and training strategies that respect the distinctive statistics of human motion without sacrificing the reasoning benefits of language models.

Tokenizing motion into a fixed codebook, typically via VQ-based models, facilitates integration with Transformer-based LLMs (Guo et al., 2022c; Zhang et al., 2023b; 2024c), however inevitably introduces quantization error, attenuating high-frequency components and degrading semantic-physical consistency. More importantly, treating motion as "language" (Jiang et al., 2023; Wang et al., 2023; Siyao et al., 2022) overlooks the gap between symbolic sequences and continuous trajectories (Wang et al., 2025b). Consequently, cross-modal alignment often remains at a symbolic level and struggles to capture the fine-grained kinematics demanded by nuanced linguistic semantics. In practice, limited codebook capacity and training coverage further constrain realism and controllability.

Recent MLLMs tend to process multiple modalities within a single backbone and attach modality-specific heads and supervision (Park et al., 2025; Team, 2024; Zhang et al., 2024b; Zhou et al., 2024). However, jointly optimizing multimodal objectives induces gradient interference and loss-scale

mismatch, which increases hyperparameter sensitivity, destabilizes training, and can erode language competence (Driess et al., 2023; Kendall et al., 2018; Tsimpoukelli et al., 2021). Moreover, forcing distinct modalities into a shared space erodes modality-specific information and inductive biases, causing negative transfer. For **robust, controllable motion–language modeling**, a method is needed that (i) adopts representations respect the continuous nature of human movement and (ii) explicitly balances multimodal, multi-objective training. Addressing these representational and optimization bottlenecks is, therefore, key to advancing unified motion understanding and generation.

**Continuous Motion Latent Space** Our approach first replace the motion tokens with a continuous, low-dimensional latent representation learned by a pretrained motion VAE (Xin et al., 2023). By 'continuous' we mean: (i) the latents are real-valued vectors rather than discrete code indices, and (ii) the VAE induces a smooth latent manifold in which nearby points correspond to gradually varying motions. Compared with VQ-based tokenization, this latent space is perceptually aligned with the original trajectories yet computationally compact, avoiding quantization artifacts and preserves high-frequency micro-dynamics for efficient and stable motion synthesis.

**Diffusion Bridge within the LLM Framework** Autoregressive generation and cross-entropy–based supervision in LLMs presumes discrete token targets and is therefore ill-suited to continuous motion latents. Conditioned on the LLM's hidden states, we further attach a lightweight diffusion head that perform denoising directly in the motion latent space to predict motion VAE latents, which the motion decoder then converts into motion sequences. Operating in a low-dimensional latent domain with a relatively small expert, this diffusion scheme bridges the gap between LLM hidden states and motion latents while only brings little overhead in both training and inference.

**Bimodal Architecture** Following Mixture-of-Transformers (MoT) (Liang et al., 2024), we treat human motion as a second modality and introduce a motion branch symmetric to the language backbone. The two independent branches interact via shared attention layers, yet retain modality-specific embeddings and allow each module to be guided by its own objective. This bimodal design mitigates interference between modalities and preserves each modality's structure, thereby enabling high-quality motion understanding and generation within a unified framework.

**Three-Stage Training** To effectively model motion branch under guidance of a pre-trained language model, we design a three-stage training scheme. First, we perform Uni-task Pretraining. with the text branch frozen, the motion branch is pre-trained on text-to-motion generation. Next, in Cross-Modal Alignment, motion-to-text and motion prediction objectives are introduced to align two branches. Finally, all parameters are optimized in Joint Fine-Tuning.

We summarize our contributions as follows:

- Latent diffusion for motion. Unlike quantization-based pipelines (Zhang et al., 2023b; 2024c), we integrate latent diffusion (Rombach et al., 2022; Xin et al., 2023) into autoregressive backbone via a diffusion head, bridging the continuous motion with the next-token prediction framework for higher-fidelity and diverse synthesis.
- Architecture and training. We propose a bimodal motion-language framework with per-modal branches communicating through shared attention, reducing interference while preserving modality-specific intelligence. A three-stage generate-then-align scheme further stabilizes joint training and curbs negative transfer.
- Results and efficiency. Under comparable settings, MotionGPT3 achieves state-of-the-art performance on both text-to-motion and motion-to-text tasks, within a unified framework, while reducing training time by approximately 2–3×.

## 2    RELATED WORK

**Human Motion Modeling** Early approaches leverage strong text encoders (Li et al., 2022; Radford et al., 2021; Raffel et al., 2020; Sanh et al., 2019) to develop motion–language understanding/retrieval via shared embeddings (Guo et al., 2022c; Tevet et al., 2022a; Yin et al., 2024) or contrastive learning (Chen et al., 2024; Petrovich et al., 2023). Recent methods (Athanasiou et al., 2024; Cohan et al., 2024; Shafir et al., 2023; Tevet et al., 2022b; Xin et al., 2023; Zhang et al., 2023c; 2024b) advance text-to-motion generation with diffusion backbones (Ho et al., 2020; Song et al., 2020), either operating directly on raw motion sequence or reconstructing a VAE latent. Working in a compressed latent space, LDMs (Rombach et al., 2022) keeps training computationally cheaper and inference faster

Figure 1: MotionGPT3 introduces hybrid motion-language model that takes motion as a second modality and processes the data through a new branch, with cross-modal attention mechanism to communicate with text branch (Sec. 3.2). We leverage a VAE network for continuous motion representation (Sec. 3.1), and design separate training objective for each modality (Sec. 3.3).

while maintaining synthesis quality. In parallel, to fit next-token–prediction recipes in large language model (LLM), several works discretize motion with VQ-VAE (Esser et al., 2021; Van Den Oord et al., 2017) into token indices, enabling transformer-based generation (Zhang et al., 2023b; 2024c; 2025). However, quantization induces approximation error and a "symbolic–continuous mismatch" that attenuates fine-grained kinematics and limits controllability, while refined tokenization such as residual VQ (RVQ) (Guo et al., 2024) and post-training schemes (Wang et al., 2025b) only partially allieviate these issues and cannot fundamentally avoid the numerical and semantic discontinuities induced by tokenization. Accordingly, we adopt an approach that interface language models directly with unquantized VAE latent representations.

Human motion modeling has evolved from task-specific designs to **unified frameworks for multi-modal understanding and generation**. Recent motion models (Jiang et al., 2023; Park et al., 2025; Wang et al., 2024; Wu et al., 2025) adopt single-stream LM backbones with discretized motion tokens to support bidirectional text-motion mapping, and have been attended to other modalities such as music (Luo et al., 2024; You et al., 2024). In parallel, language-centric multimodal frameworks extend LLMs beyond text via lightweight adapters and cross-attention conditioning, offering a generic recipe for vision- and audio-grounded reasoning (Alayrac et al., 2022; Li et al., 2023; Copet et al., 2023; Liu et al., 2023a). Works such as NExT-GPT (Wu et al., 2024b) and Janus (Wu et al., 2024a; Ma et al., 2024) employ pretrained encoders/decoders to map inputs into modality-specific latent spaces and attach adapters for flexible multimodal generation. In vision-language, Chameleon (Team, 2024) and Transfusion (Zhou et al., 2024) discretize or encode images into token sequences to support interleaved text-image training, while Show-o (Xie et al., 2024) and Fuyu (Bavishi et al., 2023) employ masked or causal attention for joint reasoning. Despite their versatility, single-stream architectures often suffer from cross-modal interference, limiting scalability and robustness. Even with carefully tuned objectives, newly introduced modalities can disrupt existing representations, underscoring the challenge of preserving modaliti-specific capability while scaling to new domains.

**Mixture-of-Experts and Multi-Stream Architecture** address these limitations by routing inputs to modality-specific experts while maintaining a shared fusion interface (Alayrac et al., 2022; Li et al., 2023; Tsimpoukelli et al., 2021). This separation reduces gradient interference between modalities, and enables branch to be guided by its own objective (Liu et al., 2021; Sener & Koltun, 2018), and simplifies the introduction of new modalities. These insights motivate hybrid strategies (Cho et al., 2024; Shi et al., 2025; Wang et al., 2025a) that combine discrete and continuous representations and decouple modality-specific encoders with *minimal* modification on the LLM backbone, thereby enhancing alignment and expressiveness. Mixture-of-Transformers (MoT) (Liang et al., 2024) instantiates this idea with modality-specific Transformer experts coupled through shared attention, facilitating modular training and reducing interference when incorporating new modalities. Guided by these observations, we adopt a MoT-style architecture that isolates motion representation learning while leveraging the language competence of pretrained LMs (Bai et al., 2023; Radford et al., 2019).

# 3 METHOD

To couple motion understanding and generation into language-centric LLMs, we observe that although discretization in prior unified systems (Jiang et al., 2023; Wang et al., 2024; Wu et al., 2025) facilitates reuse of text-style training and inference pipelines, it inevitably removes fine-grained details and complicates optimization. Moreover, single-stream backbones exacerbate cross-modal interference and yield imbalanced training. We circumvent these limitations by representing motion in a continuous, perceptually faithful VAE latent space (Sec. 3.1) and adopting a hybrid motion–text backbone that processes the two streams separately while permitting controlled interaction via shared self-attention (Sec. 3.2). On top of this backbone, we attach a diffusion head conditioned on LLM hidden states to bridge language and motion latents, enabling bidirectional understanding and generation (Sec. 3.3). Finally, together with a three-stage training schedule (Sec. 3.4), our MotionGPT3 avoids quantization bottlenecks and improves training and inference efficiency while maintaining generation quality.

## 3.1 MOTION REPRESENTATION IN CONTINUOUS TOKENS

To align motion with the autoregressive generation paradigm of large language models (LLMs) (Bai et al., 2023; Radford et al., 2019; Raffel et al., 2020; Touvron et al., 2023), previous approaches typically *descretizes* motion with vector-quantized autoencoders (Esser et al., 2021; Van Den Oord et al., 2017), converting a $N$-length sequences into latents $y \in \mathbb{R}^{n \times d}$ and replace each vector $y^i \in \mathbb{R}^d$ by its nearest code $e_k$ from a learned $K$-entry codebook. The corresponding indexes $k^{i...n}$ then serve as tokens for LLM training (Zhang et al., 2023b; Jiang et al., 2023; Wang et al., 2024).While they integrate cleanly with standard next-token prediction objectives in LLMs such as cross-entropy, quantization process inevitably introduces approximation error and disrupts motion continuity, weakening fine-grained dynamics and constraining controllability. Guo et al. (2024), equiped with residual vector quantization (RVQ), leverages multiple codebooks whose decoded latents are summed to reduce information loss. In parallel, Wang et al. (2025b) explores refined post-training tokenization. However, neither strategy fundamentally resolves the numerical and semantic discontinuities inherent to discretization.

In contrast, we adopt a continuous latent space learned by a motion VAE. Given a $N$ frame motion sequence $m^{1:N}$, the encoder $\mathcal{E}$ map $m$ into a compact continuous latent vector $z \in \mathbb{R}^d$, and the decoder $\mathcal{D}$ reconstructs $m^{1:M} = \mathcal{D}(z) = \mathcal{D}(\mathcal{E}(m^{1:M})$. The VAE is trained once with a reconstruction term (optionally including kinematic losses on pose and velocity) and a KL regularizer (Kullback & Leibler, 1951) to prevent high-variance latents and promote a smooth manifold. This compressed, continuous representation learns the inherent structure of $z$, and preserves subtle variations and maintains numerical and semantic continuity, while providing a compact domain in which our downstream generator operates. Further details can be found in the supplement.

## 3.2 BIMODAL MOTION-LANGUAGE FRAMEWORK

To accommodate the distinct characteristics of language and motion while enabling efficient cross-modal interaction, we augment a decoder-only transformer backbone Radford et al. (2019) with a *parallel* motion branch. Unlike single-stream designs that merge all modalities into one pathway Jiang et al. (2023); Wu et al. (2025), our architecture preserves modality-specific routes: a text branch $\mathcal{T}$ and a motion branch $\mathcal{M}$. Each branch maintains its own embeddings, feed-forward blocks, and normalization, and information exchange occurs only in shared self-attention layers (Alayrac et al., 2022; Shi et al., 2025). The motion branch is initialized from scratch and trained primarily under its own objective, thereby capturing motion-specific inductive biases and reducing cross-modal interference during multimodal training (Yu et al., 2020; Zhou et al., 2023).

**Hybrid Sequence Route** As illustrated in Fig. 1, given an input sequence $S = s^{1:k}$, each element is embedded either as a text embedding $\tau_i$ or as a motion latent $z_i$, with a routing indicator $\vartheta_i \in \{0, 1\}$ dispatches them to $\mathcal{T}$ or $\mathcal{M}$. The branches compute hidden states $h_t$ and $h_m$ seperately, which are then reassembled in input order for shared self-attention layers. This hybrid routing supports interleaved text–motion processing without collapsing modalities into a single embedding space, laying the foundation for high-quality, condition-aware generation.

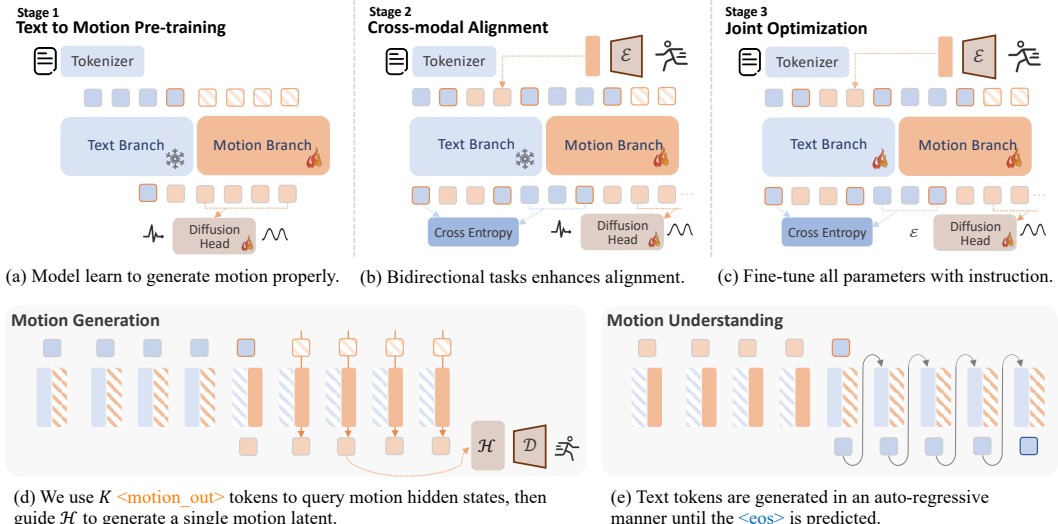

(a) Model learn to generate motion properly. (b) Bidirectional tasks enhances alignment. (c) Fine-tune all parameters with instruction.

(d) We use $K$ <motion_out> tokens to query motion hidden states, then guide $\mathcal{H}$ to generate a single motion latent.

(e) Text tokens are generated in an auto-regressive manner until the <eos> is predicted.

Figure 2: We propose a **three-stage alignment** strategy for our hybrid motion-language model: (a) The text branch is frozen, and only motion output is supervised. (b) Motion reasoning is introduced to further align the motion branch with language, with supervision on both modalities. (c) All modules are jointly fine-tuned with text branch unfrozen. (d)(e) shows inference time behavior of two branches which process data only with the same modality tags (differtiated by colors). Each rectangle block represents for a whole text/motion branch, while shadowed ones denote inactive modules. Modalities are color-coded: blue for text and orange for motion. Shadowed orange squares represent <motion_out>, and orange-outlined squares indicate boundary tokens <som> or <eom>.

**Interfaces for Continuous Motion Latents** Because that the continuous motion representation (Xin et al., 2023) does not rely on a tokenized vocabulary or codebook, the index-to-embedding lookup and softmax decoding employed for text cannot be reused for motion. We therefore introduce motion-specific interfaces that bridge continuous latents and transformer hidden states. First, we augment the text vocabulary with a small set of motion-boundary/holder tokens (i.e. <som> <eom>, <motion_in>, <motion_out>) to mark motion spans and I/O positions in interleaved sequences as in Zhou et al. (2024). Second, a Motion Understanding Head (MUH) linearly maps motion latents into the Transformer's input embedding space for captioning and prediction. Finally, a lightweight Motion Generation Head (MGH) projects hidden states back to the VAE latent space via diffusion (Ho et al., 2020; Rombach et al., 2022; Xin et al., 2023).

### 3.3 MOTION DIFFUSION IN AUTOREGRESSIVE BACKBONE

Continuous representations are inherently misaligned with the discrete nature of token-based generation in LLMs, and requires more sophisticated modeling to support generation. Inspired by recent advances in diffusion-based generative modeling (Tevet et al., 2022b; Song et al., 2020; Rombach et al., 2022; Li et al., 2024a), we attach a lightweight diffusion module $\mathcal{H}$ to bridge this gap. $\mathcal{H}$ predicts motion latents directly from the backbone's hidden states, enabling integration of continuous motion representation within an autoregressive framework.

**Diffusion Process** Given a ground-truth motion sequence $x$, we obtain the target latent $z_0 = \mathcal{E}(x) \in \mathbb{R}^d$ via the motion encoder $\mathcal{E}$. We adopt a fixed forward noising process over $t \in \{1, \ldots, T\}$ with Gaussian perturbations: $z_t = \sqrt{\bar{\alpha}_t}\, z_0 + \sqrt{1 - \bar{\alpha}_t}\, \epsilon$, where $\epsilon \sim \mathcal{N}(0, I)$ and $\bar{\alpha}_t = \prod_{s=1}^{t} \alpha_s$ is the cumulative product of noise scheduling coefficients. A time-aware denoiser $\mathcal{H}$ is conditioned on the Transformer hidden states from the motion stream (denoted $h_m$) and learns to reverse the diffusion process. Conditioning is implemented via lightweight linear projections into the denoiser inputs, following recent practice (Ho & Salimans, 2022; Li et al., 2024a). We train $\mathcal{H}$ with the standard DDPM objective (Ho et al., 2020; Song et al., 2020): $\mathcal{L}_{\text{diff}} = \mathbb{E}_{z_0,\, t,\, \epsilon}\left[\left\| \epsilon - \mathcal{H}(z_t, t, h_m) \right\|_2^2 \right].$

**Inference** The text branch autoregressively generates tokens until a motion start marker <som> is produced. We then insert $K$ placeholder tokens (i.e., <motion_out>) to elicit the span-aligned hidden states $h_m^{i:i+K}$ in a single forward pass. The diffusion head runs the reverse process conditioned on $h^{i:i+K}$ to sample the noise-free motion latent $\hat{z}_0$, which the VAE decoder $\mathcal{D}$ finally decoded to the raw motion sequence. As in Team (2024), generation then resumes in the text stream with a concatenated <eom> until end-of-sequence. Operating on compact motion latents, this diffusion head adds only minimal overhead during training and inference.

### 3.4 TRAINING PROCEDURE

We first train a motion VAE to obtain a compact, continuous latent space, following prior works (Rombach et al., 2022; Xin et al., 2023). The bimodal backbone then adopts a pretrained decoder-only LLM (Radford et al., 2019) as the text branch $\mathcal{T}$. As described in Fig. 2, the motion branch $\mathcal{M}$ is initialized from scratch and brought into alignment with $\mathcal{T}$ through a three-stage schedule.

**Stage I: Text-to-motion pretraining** We begin by pretraining $\mathcal{M}$ on text-to-motion, while freezing $\mathcal{T}$. This provides stable linguistic conditioning and biases the model toward motion-specific semantics. In this stage, $\mathcal{M}$ conditions on the frozen language representations and is trained via diffusion, to synthesize VAE motion latentss, where diverse text–motion pairs encourages a rich and flexible mapping from language to the latent space (Rombach et al., 2022; Xin et al., 2023).

**Stage II: Cross-Modal Alignment** Keeping $\mathcal{T}$ frozen, we introduce additional objectives to couple understanding and generation. Concretely, training includes multiple tasks of text-to-motion (T2M), motion-to-text (M2T), and motion prediction. Following instruction-style formulations in Jiang et al. (2023), these tasks are further presented as prompts covering generation, captioning, prediction, and inbetweening. Multi-task optimization fosters bidirectional alignment without forcing a single shared representation and encourages motion representations that are semantically coherent with language features (Alayrac et al., 2022; Li et al., 2023).

**Stage III: Joint Fine-Tuning** Finally, we unfreeze $\mathcal{T}$ and fine-tune all parameters via instruction tuning on a mixture of paired text–motion data and, optionally, text-only prompts (Dai et al., 2023; Liu et al., 2023b; Wei et al., 2021). Including text-only prompts can further improve language competence for downstream applications.

## 4 EXPERIMENTS

We empirically validate MotionGPT3, a dual-stream architecture for efficient, language-grounded multimodal motion understanding and generation, across motion-centric tasks. Dataset configurations, evaluation metrics, and implementation details are summarized in Sec. 4.1. Begin with analyzing optimization dynamics and inference efficiency via training loss and validation curves (Sec. 4.2), we then present controlled ablations that isolate the contributions of the continuous VAE motion representation and the bimodal design (Sec. 4.4). Next, We benchmark MotionGPT3 on text-to-motion generation and motion-to-text understanding, comparing against both specialized single-task methods and unified state-of-the-art systems (Sec. 4.3). Finally, we ablate the proposed three-stage training scheme (Sec. 4.4). Additional qualitative results are provided in the supplement.

### 4.1 EXPERIMENTAL SETUP

**Datasets** We train and evaluate our model mainly on HumanML3D (Guo et al., 2022b), alarge-scale benchmarks for text-motion generation and understanding, and KIT-ML (Plappert et al., 2016) with 3,911 motionsequences. For comparison with prior works (Xin et al., 2023; Jiang et al., 2023), we adopt the 263-dim pose proposed in Guo et al. (2022b), which combines joint velocities/ positions/ rotations, and foot-contact signals, following the standard data split.

**Evaluation Metrics** We evaluate two tasks. For the *text-to-motion*, we follow the previous works (Guo et al., 2022c; Jiang et al., 2023; Xin et al., 2023; Zhang et al., 2023b) to report motion quality (FID), diversity (DIV and MM), and text-motion alignment (R-Precision and MMDist). For *motion-to-text*, we use both alignment metrics (R-Precision and MMDist) and linguistic metrics from NLP (Bleu (Papineni et al., 2002), Rouge-L (Lin, 2004), CIDEr (Vedantam et al., 2015), and BertScore (Zhang et al., 2019)). See Sec. D.3 for metric definitions and computation details.

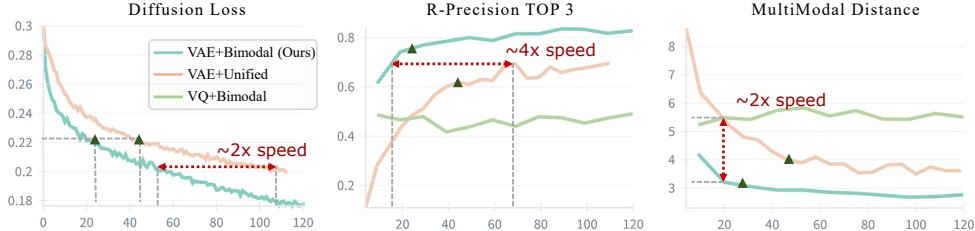

Figure 3: Training loss and validation curves on motion generation on HumanML3D for architecture variants of dual-stream and single-stream and representation variants of VAE and VQ latents. The right figures illustrate validation metrics of R-Precision TOP 3 (R@3↑) and Multimodal Distance (MMDist↓). Triangle markers indicate matched-loss checkpoints (∼0.22). Our hybrid architecture with continuous motion representation helps accelerating convergence for about 2×, as well as achieves better quality especially in early training stage.

**Implementation Details** Our framework comprises three main components: a motion VAE, a lightweight diffusion head, and a dual-stream backbone. We adopt the Transformer-based motion VAE of Xin et al. (2023), where both encoder and decoder consists of 9 layers and 4 heads with skip connections, producing a $1 \times 1 \times 256$ latent per motion sequence. Our Diffusion Head $\mathcal{H}$ is implemented as a 3-layer MLP with ResBlock-style layers and hidden dimension 1024, following Li et al. (2024a). We train diffusion with a scaled linear noise schedule for 1000 denoising steps, while inference uses 100 steps by default. The text and motion branches share the GPT-2 base configuration but use disjoint parameters, both are decoder-only with 12 Transformer layers, model dimension 768, and MLP dimension 3072, unless stated otherwise. The text branch is initialized from a pretrained 124M GPT-2 checkpoint, while the motion branch from scratch, yielding total 238M parameters.

**Training Protocol** We use AdamW for all components, with a learning rate of $1 \times 10^{-4}$ for the motion backbone and $2 \times 10^{-4}$ for the diffusion head. Training uses a mini-batch size of 32 on 2 NVIDIA RTX 3090 GPUs, with identical training/inference settings on HumanML3D (Guo et al., 2022b). The motion VAE is trained with a learning rate of $1 \times 10^{-4}$, batch size 256, over 150K iterations. The motion-language backbone is trained for 100k iterations in text-to-motion pretraining, followed by 300k iterations for cross-modal alignment.

## 4.2 TRAINING EFFICIENCY WITH BIMODAL ARCHITECTURE AND CONTINUOUS LATENTS

To assess our two main design choices: (i) the dual-stream (bimodal) backbone and (ii) continuous representation from motion VAE, we analyze training dynamics on the text-to-motion task under identical settings on HumanML3D. Fig. 3 plots training loss and validation metrics over time for three variants of single-stream+VAE, dual-stream+VAE, and dual-stream+VQ.

We observe that i) **Discrete VQ latents plateau at a lower quality ceiling.** The VQ baseline (VQ + bimodal; green curve) reaches ≈0.5 R-Precision Top 3 (R@3) early in training and then saturates, yielding substantially lower R@3 and higher MultiModal Distance (MMDist) than the VAE-based counterparts. This likely stems from quantization-induced information loss (Guo et al., 2024; Wang et al., 2025a) and tokenization that disrupts the semantic continuity of motion. ii) **Bimodal design accelerates optimization.** Compared with single-stream backbone, the dual-stream variant reduces diffusion loss roughly 2× faster and sustains superior validation performance over the entire training trajectory, in terms of R@3 and MMDist. iii) **Superior quality at matched optimization states.** At comparable diffusion loss, the dual-stream model still leads. For instance, we mark by triangles a loss of ≈0.22, reached at ∼20 epochs for dual-stream model and ∼40 epochs for the single-stream model, the former achieves higher R@3 and lower MMDist.

We attribute these effects to **modality-aware optimization**: the motion branch learns motion-specific semantics while the language branch focuses on textual cues. Decoupling the streams and supervising them separately mitigates gradient interference and avoids the representational compromises common in single-stream models (Liu et al., 2021; Sener & Koltun, 2018; Yu et al., 2020). Although a shared space might be expected to bring modalities "closer", in practice single-stream coupling can entangle modality structure and yield counterintuitive outcomes. In summary, continuous VAE latents within a dual-stream backbone strike a favorable efficiency–fidelity trade-off, enabling high-quality motion synthesis with reduced training time.

## 4.3 COMPARISONS

By modeling human motion as a second modality alongside language, our bimodal motion-language model supports both text-to-motion (T2M) and motion-to-text (M2T). We report results for two settings: (i) single-task models trained specifically for target task (MotionGPT3$^\dagger$) , and (ii) a unified model (MotionGPT3) trained on both tasks with the three-stage scheme .

**Text-to-Motion Generation**  The text-to-motion task involves generating realistic and diverse motion sequences conditioned on natural language descriptions. We evaluate a single-task generator MotionGPT3† as well as a multi-task generator MotionGPT3. We compare MotionGPT3† against recent methods (Guo et al., 2022b;c; Lou et al., 2023; Jiang et al., 2023; Xin et al., 2023; Zhang et al., 2023b;c; Guo et al., 2024; Wu et al., 2024c; Li et al., 2024b; Zhang et al., 2025) and evaluate the performance on both HumanML3D and KIT-ML datasets. Following Guo et al. (2022b), each evaluation is repeated 20 times and reported with 95% confidence intervals. As shown in Tab. 1 and Tab. 2, MotionGPT3† matches or exceeds generation-only baselines (Zhang et al., 2023b; Lou et al., 2023; Guo et al., 2024) on alignment metrics (R@k, MMDist), with competitive FID and diversity. Further, the unified MotionGPT3performs better than recent unified systems (Guo et al., 2022c; Jiang et al., 2023; Wu et al., 2024c; 2025). See comaprison with more approaches in Sec. C.1.

**Motion-to-Text Understanding**  The motion-to-text task involves understanding motion sequences and generating semantically appropriate textual descriptions. We train a single-task captioner MotionGPT3$^\dagger$ for 100 epochs and compare with recent SOTA (Guo et al., 2022c; Jiang et al., 2023; Li et al., 2024b; Wu et al., 2024c). Following Jiang et al. (2023), we evaluate on the raw ground truth texts using the TM2T protocol (Guo et al., 2022c). Results in Tab. 3 show that both MotionGPT3$^\dagger$ and MotionGPT3  achieve strong retrieval performance and language metrics. Notably, we observe a marked reduction in Multimodal Distance (MMDist), indicating effective motion–language alignment under the dual-stream design.

Table 1: Evaluation of text-guided motion generation on HumanML3D  (Guo et al., 2022a). Rows are grouped by training tasks: *Gen. only* for generation-only and *Gen. & Und.* for both. *Real* is obtained by ground-truch motions, and → indicate values closer to *Real* are desirable. † marks our single-task model trained for 200 epochs, and MotionGPT3 is a three-stage model trained with unified tasks. Best and second-best results are highlighted in **bold** and underline.

| Types | Methods | R@1 | R@2 | R@3 | FID↓ | MMDist↓ | Diversity→ | MModality↑ |
|---|---|---|---|---|---|---|---|---|
| | Real | 0.511 | 0.703 | 0.797 | 0.002 | 2.974 | 9.503 | - |
| Gen. only | T2M-GPT Zhang et al. (2023b) | 0.491 | 0.680 | 0.775 | 0.116 | 3.118 | 9.761 | 1.856 |
| | DiverseMotion Lou et al. (2023) | 0.515 | 0.706 | 0.802 | **0.072** | 2.941 | 9.683 | 1.869 |
| | MotionDiffuse Zhang et al. (2024a) | 0.491 | 0.681 | 0.782 | 0.630 | 3.113 | 9.410 | 1.553 |
| | MoMask Guo et al. (2024) | 0.521 | 0.713 | 0.807 | 0.045 | 2.958 | 9.620 | 1.241 |
| | **MotionGPT3**† | *0.533* | 0.731 | 0.826 | 0.239 | 2.797 | 9.688 | 1.560 |
| Gen. & Und. | TM2T Guo et al. (2022c) | 0.424 | 0.618 | 0.729 | 1.501 | 3.467 | 8.589 | **2.424** |
| | MotionGPT Jiang et al. (2023) | 0.492 | 0.681 | 0.733 | 0.232 | 3.096 | **9.528** | 2.00 |
| | MoTe Wu et al. (2024c) | 0.548 | 0.737 | 0.825 | 0.075 | 2.867 | - | *2.399* |
| | MG-MotionLLM Wu et al. (2025) | 0.516 | 0.706 | 0.802 | 0.303 | 2.952 | 9.960 | 2.125 |
| | **MotionGPT3** | **0.553** | **0.747** | **0.837** | 0.208 | **2.725** | 9.700 | 1.018 |

Table 2: Evaluation of text-guided motion generation on KIT-ML  (Plappert et al., 2016).

| Methods | R@1 | R@2 | R@3 | FID↓ | MMDist↓ | Diversity→ | MModality↑ |
|---|---|---|---|---|---|---|---|
| Real | 0.424 | 0.649 | 0.779 | 0.031 | 2.788 | 11.08 | - |
| MLD Xin et al. (2023) | 0.390 | 0.609 | 0.734 | 0.404 | 3.204 | 1.080 | 2.192 |
| MotionGPT Jiang et al. (2023) | 0.366 | 0.558 | 0.680 | 0.510 | 3.527 | 10.350 | 2.328 |
| MotionDiffuse Zhang et al. (2024a) | 0.417 | 0.621 | 0.739 | 1.954 | 2.958 | **11.100** | 0.730 |
| ReMoDiffuse Zhang et al. (2023c) | 0.427 | 0.641 | 0.765 | 0.155 | 2.814 | 10.800 | 1.239 |
| MoMask Guo et al. (2024) | 0.433 | 0.656 | 0.781 | 0.204 | 2.779 | - | 1.131 |
| MoTe Wu et al. (2024c) | 0.419 | 0.627 | 0.741 | 0.256 | 3.216 | - | **2.615** |
| MoGenTS Yuan et al. (2024) | 0.445 | 0.671 | 0.797 | **0.143** | 2.711 | 10.918 | - |
| **MotionGPT3**† | **0.456** | **0.680** | **0.803** | 0.227 | **2.704** | 11.026 | 0.9036 |

Table 3: Comparison of motion captioning on HumanML3D (Guo et al., 2022a), evaluation follows (Guo et al., 2022c). MotionGPT3 † denotes our single-task captioning model trained for 100 epochs, and MotionGPT3 is an unified model trained on both tasks with the three-stage scheme (Sec. 3.4). Both variants achieve R@k on par with recent state of the art, and surpass the GT metrics.

| Methods | R@1 | R@2 | R@3 | MMDist ↓ | Bleu@1 ↑ | Bleu@4 ↑ | Rouge ↑ | Cider ↑ | BertScore ↑ |
|---|---|---|---|---|---|---|---|---|---|
| Real | 0.523 | 0.725 | 0.828 | 2.901 | - | - | - | - | - |
| TM2T (Guo et al., 2022c) | 0.516 | - | 0.823 | 2.935 | 48.9 | 7.00 | 38.1 | 16.8 | 32.2 |
| MotionGPT (Jiang et al., 2023) | 0.543 | - | 0.827 | 2.821 | 48.2 | 12.5 | 37.4 | 29.2 | 32.4 |
| LaMPM2T (Li et al., 2024b) | 0.547 | - | 0.831 | 2.808 | 47.8 | 13.04 | 37.1 | 28.9 | 32.7 |
| MoTe (Wu et al., 2024c) | **0.577** | - | **0.871** | 2.649 | 46.7 | 11.15 | 37.4 | **31.5** | 30.3 |
| **MotionGPT3**† | 0.553 | 0.756 | 0.853 | 2.524 | 56.363 | 17.661 | 44.997 | 30.980 | **35.850** |
| **MotionGPT3** | 0.573 | **0.773** | 0.864 | **2.426** | **59.083** | **19.412** | **46.173** | 28.721 | 35.231 |

## 4.4 ABLATION STUDIES

This section reports quantitative ablations. In contrast to training-curve analysis in Sec. 4.2, we evaluate final test-set performance on both text-to-motion (T2M) and motion-to-text (M2T). First, we assess the contributions of a dual-stream backbone and continuous VAE motion latents by varying one factor at a time. Then, we analyze the proposed three-stage training schedule and quantify its effects We also examine the impact of hidden-state processing in the Diffusion Head $\mathcal{H}$ and the use of classifier-free guidance (CFG). See Sec. C for more detailed experiments.

**Model Design** Tab. 4 summarized test-set results on HumanML3D (Guo et al., 2022a). We evaluate T2M and M2T separately and compare four variants obtained by crossing architecture (single- vs. dual-stream) with representation (discrete VQ vs. continuous VAE). Under the same evaluation protocol, replacing VAE with VQ or replacing a dual-stream backbone (Bimodal) with a single-stream one (Unified) consistently degrades performance on both tasks. Notably, changing the architecture change to Bimodal yields larger gains on M2T, whereas changing the representation to VAE yields larger gains on T2M. This task-dependent sensitivity is consistent with Sec. 4.2: decoupling streams mitigates cross-modal interference and benefits semantic-level alignment, while continuous latents reduce quantization loss and improve synthesis fidelity for motion generation.

Table 4: Component ablations on HumanML3D for representation choice and architecture design. *Unified* denotes a single-stream backbone, where one branch is shared by text and motion, as employed in Jiang et al. (2023), and *Bimodal* denotes a dual-stream backbone described in Sec. 3.2. VQ and VAE indicate discrete and continuous motion latents, respectively. For each configuration we train separate models for motion generation (T2M) and motion captioning (M2T) under the same protocol and report test-set metrics. All variants share the same GPT-2-style branch and hyperparameters. and training is run for 100 epochs on M2T and 200 epochs on T2M. Best and second-best results are highlighted in **bold** and underline.

| Settings | Text-to-Motion | | | | Motion-to-Text | | | |
|---|---|---|---|---|---|---|---|---|
| | R@1 ↑ | R@3 ↑ | MMDist ↓ | FID ↓ | R@1 ↑ | R@3 ↑ | MMDist ↓ | BertScore ↑↑ |
| Real | $0.511^{\pm0.003}$ | $0.797^{\pm0.002}$ | $2.974^{\pm0.008}$ | $0.002^{\pm0}$ | 0.523 | 0.828 | 2.901 | - |
| Unified+VQ | $0.237^{\pm0.003}$ | $0.435^{\pm0.003}$ | $5.684^{\pm0.018}$ | $0.403^{\pm0.014}$ | - | - | - | - |
| Unified+VAE | $0.501^{\pm0.003}$ | $0.792^{\pm0.002}$ | $2.841^{\pm0.011}$ | $0.489^{\pm0.017}$ | 0.234 | 0.426 | 5.976 | 16.197 |
| Bimodal+VQ | $0.300^{\pm0.005}$ | $0.532^{\pm0.02}$ | $4.937^{\pm0.077}$ | $0.454^{\pm0.078}$ | 0.379 | 0.702 | 3.545 | 18.085 |
| Bimodal+VAE | **$0.533^{\pm0.002}$** | **$0.826^{\pm0.003}$** | **$2.797^{\pm0.007}$** | **$0.239^{\pm0.008}$** | **0.553** | **0.853** | **2.524** | **35.850** |

T2M with *Bimodal+VQ* and *Unified+VQ* is extended to 400 epochs to approach convergence.
M2T results for *Unified+VQ* is not reported are omitted because performance remained unevaluable after more than 400 training epochs.

**Cross-Modal Attention** We further enable CMA in the last $L$ layers and sweep $L$ (Fig. 4). Increasing $L$ from 2 to 5 generally improves quality, reflected by lower FID scores and MMDist and higher R-Precision. However, the trend is **non-monotonic**: we observe a slight drop at $L = 6$ in R-Precision, relative to $L = 5$, may suggesting that late but not ubiquitous CMA is preferable. More results on CMA schedules can be found in Sec. C.3.

**Training Stage** We ablate the three-stage schedule in Sec. 3.4, including 100k iters on text-to-motion pretraining (SI), 300k iters on cross-modal alignment (SII), and 50k iters on joint fine-tuning (SIII), and evaluate on both T2M and M2T. Results are summarized in Tab. 5. SI already yields strong generation and provides a motion-specialized initialization. Optimization in SII confers

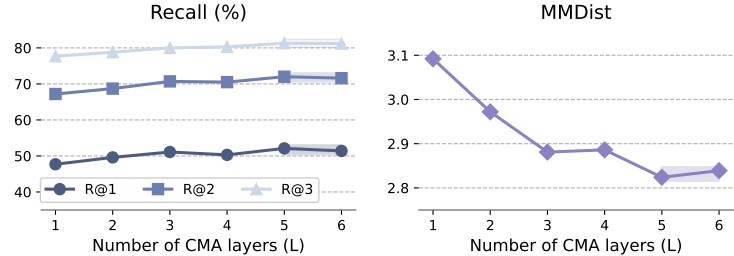

Figure 4: Ablation on the number of cross-modal attention (CMA) layers for T2M on HumanML3D. CMA is enabled in the last $L$ layers ($L \in 1, \ldots, 6$). Performance improves as $L$ increases up to 5 layers, then shows slight degradation at 6, indicating a **non-monotonically pattern**.

Table 5: Ablation on training-scheme. Enabled stages are marked with ✓, and colors encode the text branch updated or **frozen**. Best results are **bold** and second best are underlined.

| Stage I | Stage II | Stage III | Text-to-Motion | | | Motion-to-Text | | |
|---|---|---|---|---|---|---|---|---|
| | | | R TOP3 ↑ | FID ↓ | MMDist ↓ | R TOP1↑ | Bleu@4↑ | Bert↑ |
| ✓ | ✗ | ✗ | 0.826 | 0.239 | 2.797 | - | - | - |
| ✓ | ✓ | ✗ | 0.831 | 0.215 | 2.755 | 0.571 | 18.328 | 33.993 |
| ✓ | ✓ | ✓ | 0.837 | 0.208 | 2.725 | 0.573 | 19.412 | 35.231 |
| ✗ | ✓ | ✓ | 0.772 | 0.325 | 3.108 | 0.573 | 18.277 | 35.546 |

M2T capability and, importantly, further improves T2M, by $-0.10$ on FID and $-0.2$ on MMD. indicating that explicit alignment benefits both directions. Without extra text-only supervision, SIII adds small additional gains on M2T while preserving T2M, serving as a light joint refinement rather than a substitute for S2. As shown in the last row of Tab. 5, a two-stage model that *omits SI* keeps M2T largely intact but markedly degrades T2M, underscoring the role of S1 in learning motion-specific features. Overall, the full three-stage schedule provides the best trade-off, delivering reliable generation and captioning with well-aligned motion–language representations. Additional variants are reported in Sec. C.6, including experiments with an unfrozen text branch.

## 5  DISCUSSION

To address quantization-induced degradation and cross-modal interference in multi-objective training, we present **MotionGPT3**, a dual-stream motion-language framework that unifies motion understanding and generation while preserving modality-specific inductive biases. By encoding motion as continuous VAE latents and generating in latent space with a lightweight diffusion head, the model avoids quantization artifacts and improves synthesis fidelity. The dual-stream Transformer with shared attention enables controlled bidirectional exchange, which strengthens text–motion alignment, reduces cross-modal interference, and empirically accelerates single-task convergence without degrading quality. For joint training of understanding and generation, a generate-then-align three-stage training schedule further stabilizes optimization and mitigates cross-task interference.

**Limitations and Failure Cases**  Fine-grained control cam fail on directional cues (e.g., left/right). Because the current VAE yields a single latent per sequence, segment-level composition and local semantic alignment for long motions are not explicitly supported. Generalization to out-of-domain descriptions is constrained by data coverage. Potential remedies include incorporating diverse text-only corpora in the final alignment stage, adopting stronger language backbones, and exploring hierarchical or segment-wise latent representations to enable compositional control. **Future Work** We will scale training to (i) larger, more diverse datasets, (ii) develop controllable motion with local semantic alignment and segment-level for long-horizon generation, and (iii) evaluate the framework with stronger language models and larger-scale training regimes to assess efficiency and robustness.

**Acknowledgment**  This work was supported in part by NSF China (No. 62172363). This work was supported by the JC STEM Lab of AI for Science and Engineering, funded by The Hong Kong Jockey Club Charities Trust, the MTR Research Funding (MRF) Scheme (CHU-24003), the Research Grants Council of Hong Kong (Project No. CUHK14213224).

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

# APPENDIX

This appendix provides qualitative comparison results (Sec. B), disccusion of continuous/ discrete motion representation (Sec. A), additional quantitative results and ablation (Sec. C) on motion branch size and connection type, motion supervision scheme, training stages. We also provide more implementation details in Sec. D. Please note our examination of metrics report on TMR evaluator (Sec. C.2), analysis on bimodal architecture (Sec. C.3), ablation on our training scheme (Sec. C.6). and more language backbones (Sec. C.7).

**Website & Video** A supplementary website provides visualizations of quantitative results, motion data, and demonstration videos. A standalone video is also available on the website and at `supp/website_video/static/videos/MotionGPT3/_Video.mp4`, showcasing (i) text-to-motion comparisons, (ii) motion-captioning comparisons, and (iii) additional results on motion generation and captioning.

**Code** Example code files are available in the supplementary materials, which cover the training and evaluation processes of our MotionGPT3 models, along with several example results.

## A  DISCRETE TOKEN VS CONTINUOUS TOKEN

**Reconstruction Task** We compare continuous latents from MLD VAE (Xin et al., 2023) with discrete latents from VQ-VAE (Jiang et al., 2023) for motion reconstruction under identical settings. As shown in Sec. A, VQ-VAE yields higher errors on MPJPE, PAMPJPE, ACCL, and APE/AVE for root, trajectory, pose, joints, indicating reduced fidelity and temporal smoothness relative to the continuous VAE. This gap is expected: quantization maps a continuous motion manifold to a finite codebook and turns motion modeling into token classification, introducing unavoidable approximation error and information loss. In practice, many distinct frames collapse to the same code (one-to-many mapping), yielding ambiguous reconstructions and frame-wise noise that harms smoothness.

Table 4: Reconstruction performance of a continuous VAE (Xin et al., 2023) versus a discrete VQ-VAE (Jiang et al., 2023). The VQ-VAE shows consistently higher errors, consistent with information loss introduced by quantized encoding and decoding. Sec. D.3 presents the metric definitions.

| Method | MPJPE | PAMPJPE | ACCL | APE | | | | AVE | | | |
|--------|-------|---------|------|------|------|------|--------|------|------|------|--------|
| | | | | root | traj | pose | joints | root | traj | pose | joints |
| VAE | 43.906 | 31.356 | 5.93 | 0.0581 | 0.0504 | 0.0277 | 0.0619 | 0.0179 | 0.0177 | 0.0012 | 0.0185 |
| VQ | 46.828 | 33.668 | 7.629 | 0.0829 | 0.0804 | 0.0316 | 0.0930 | 0.0240 | 0.0239 | 0.0015 | 0.0253 |

**On Discrete VQ Latents** Recent work  Guo et al. (2024) addresses the limitations of codebook capacity using hierarchical Residual Vector Quantization (RVQ) with separate predictors for base and residual tokens.  Wang et al. (2025b) combines quantized and continuous latents via post-training quantization.  Cho et al. (2025) introduces a diffusion-based decoder that progressively maps discrete tokens back to continuous raw motions, improving fidelity and smoothness, and uses the symmetric Jerk Percentage Error (sJPE) to detect under-reconstruction and frame noise. Despite these advances, discrete pipelines remain prone to expressiveness bottlenecks and token-induced jitter. We further evaluate VQ and VAE latents within dual-stream architecture under single-task training. The results (Tab. 5) consistently favor continuous representations for both generation and understanding. While discrete codes facilitate token-based modeling, continuous representations better capture fine-grained dynamics and motion continuity, achieving higher alignment and synthesis quality with fewer epochs.

Table 5: Discrete (VQ) vs. continuous (VAE) motion representations under single-task training. We report both T2M on R@1, FID, MMDist, DIV and M2T on R@3, BLEU@1/4, ROUGE. With fewer training epochs, VAE-variants achieve stronger alignment and better quality. VQ requires extended training of 399 epochs while still remains behind on most alignment and language scores.

| Representation | Text-to-Motion | | | | | Motion-to-Text | | | | |
|----------------|-------|-------|----------|-------|--|-------|-------|----------|----------|--------|
| | epoch | R@1↑ | FID↓ | MM Dist↓ | DIV→ | epoch | R@3↑ | Bleu@1↑ | Bleu@4↑ | Rouge↑ |
| VQ | 199 | 0.258 | 0.542 | 5.364 | 9.274 | 99 | 0.765 | 47.043 | 7.234 | 39.244 |
| | 399 | 0.300 | 0.454 | 4.937 | 9.626 | 199 | 0.752 | 41.579 | 6.304 | 35.746 |
| VAE | 199 | 0.525 | 0.191 | 2.667 | 10.095 | 99 | 0.859 | 50.707 | 8.383 | 38.225 |

**Generation Task (T2M)** With the same 200 training epochs, the VAE representation delivers substantially better quality than VQ, with 0.525 vs. 0.258 on R@1 and 2.667 vs. 5.364 on MMDist, while maintaining competitive diversity(DIV). Extending VQ to 399 epochs reduces FID to 0.454, however, it still lags in alignment, with R@1 0.300 and MMDist 4.937, indicating a lower quality ceiling for discrete codes.

**Understanding Task (M2T)** At matched training over 99 epochs, VAE attains stronger retrieval and language scores than VQ, with 0.859 vs. 0.765 on R@3 and 50.707 vs. 47.043 on BLEU@1, while ROUGE is comparable. Prolonged VQ training to 199 epochs unexceptedly reduces performance across all language scores with R@3 dropping by 0.13 and BLEU@1 by 5.464, suggesting optimization instability and poorer generalization under the discrete setting.

## B  QUALITATIVE RESULTS

We provide qualitative comparisons for text-to-motion (T2M) Fig. 5, motion-to-text (M2T) Fig. 6, and a multi-task gallery Fig. 7. To ensure fair visualization, we use a fixed list of prompts from the HumanML3D test split; all clips are rendered at 20 FPS with identical camera and skeleton settings. Baselines are run with the their official checkpoints. Overall, MotionGPT3 exhibits stronger smoother transitions and text-motion alignment. Discrete-code variants tend to show token-induced frame noise and temporal drift, whereas single-stream models can produce semantically inconsistent motions on complex prompts. Additional examples and animations are provided in the supplementary video.

## C  ADDITIONAL EXPERIMENTS

This section provides supplementary experiments for further evaluation of MotionGPT3. First, we report a comprehensive comparison with more baselines on text-to-motion generation (Sec. C.1) and further assess text-to-motion with the TMR retrieval evaluator (Sec. C.2). We then analyze design choices of the Cross-Modal Attention (CMA) (Sec. C.3), examine scaling effects of both language and motion branches (Sec. C.4), analyze the diffusion-based supervision used for motion generation (Sec. C.5), and evaluate the effectiveness of the three-stage training strategy (Sec. C.6). Fianlly, we further provide additional experiments with more language backbones (Sec. C.7).

### C.1  QUANTITATIVE RESULTS

Tab. 6 reports the full text-to-motion results on HumanML3D, grouped by training regime (generation-only vs. unified dual-task). Notably, evaluated on the HumanML3D dataset by T2M evaluator (Guo et al., 2022b), recent models consistently achieve very high scores, and several recent approaches (Guo et al., 2024; Zhang et al., 2025; Li et al., 2024b; Wu et al., 2024c; 2025), including MotionGPT3, achieve scores even above those of the ground-truth data (*Real*).

Table 6: Comprehensive comparison of text-to-motion generation on HumanML3D (Guo et al., 2022a). We report generation-only models (Gen. only) here, and visualize unified dual-task models (Gen. & Und.) in Fig. 8. Real denotes ground-truth statistics; arrows ($\rightarrow$) indicate that values closer to Real are desirable. † marks our single-task model trained for 200 epochs, and MotionGPT3 is the unified three-stage model. Best and second-best results are **bold** and underlined.

| Types | Methods | R@1 | R@2 | R@3 | FID↓ | MMDist↓ | DIV→ | MModality↑ |
|---|---|---|---|---|---|---|---|---|
| | Real | $0.511^{\pm.003}$ | $0.703^{\pm.003}$ | $0.797^{\pm.002}$ | $0.002^{\pm0}$ | $2.974^{\pm.008}$ | $9.503^{\pm.065}$ | |
| Gen. only | T2M (Guo et al., 2022b) | $0.457^{\pm.002}$ | $0.639^{\pm.003}$ | $0.74^{\pm.003}$ | $1.067^{\pm.002}$ | $3.34^{\pm.008}$ | $9.188^{\pm.002}$ | $2.09^{\pm.083}$ |
| | MLD (Xin et al., 2023) | $0.481^{\pm.003}$ | $0.673^{\pm.003}$ | $0.772^{\pm.002}$ | $0.473^{\pm.013}$ | $3.169^{\pm.01}$ | $9.724^{\pm.082}$ | $\underline{2.413}^{\pm.079}$ |
| | MotionDiffuse (Zhang et al., 2024a) | $0.491^{\pm.001}$ | $0.681^{\pm.001}$ | $0.782^{\pm.001}$ | $0.630^{\pm.001}$ | $3.113^{\pm.001}$ | $9.410^{\pm.049}$ | $1.553^{\pm.042}$ |
| | T2M-GPT (Zhang et al., 2023b) | $0.491^{\pm.003}$ | $0.68^{\pm.003}$ | $0.775^{\pm.002}$ | $0.116^{\pm.004}$ | $3.118^{\pm.011}$ | $9.761^{\pm.081}$ | $1.856^{\pm.011}$ |
| | ReMoDiffuse (Zhang et al., 2023c) | $0.510^{\pm.005}$ | $0.698^{\pm.006}$ | $0.795^{\pm.004}$ | $0.103^{\pm.004}$ | $2.974^{\pm.016}$ | $9.018^{\pm.075}$ | $1.795^{\pm.043}$ |
| | DiverseMotion (Lou et al., 2023) | $0.515^{\pm.003}$ | $0.706^{\pm.002}$ | $0.802^{\pm.002}$ | $0.072^{\pm.004}$ | $2.941^{\pm.007}$ | $9.683^{\pm.102}$ | $1.869^{\pm.089}$ |
| | MoMask (Guo et al., 2024) | $0.521^{\pm.002}$ | $0.713^{\pm.002}$ | $0.807^{\pm.002}$ | $0.045^{\pm.002}$ | $2.958^{\pm.008}$ | $9.620^{\pm.064}$ | $1.241^{\pm.04}$ |
| | MoGenTs (Yuan et al., 2024) | $0.529^{\pm.003}$ | $0.719^{\pm.002}$ | $0.812^{\pm.002}$ | $0.033^{\pm.002}$ | $2.867^{\pm.006}$ | $9.570^{\pm.07}$ | - |
| | MotionAnything (Zhang et al., 2025) | $0.546^{\pm.003}$ | $0.735^{\pm.002}$ | $0.829^{\pm.002}$ | $\textbf{0.028}^{\pm.005}$ | $2.859^{\pm.01}$ | $\textbf{9.521}^{\pm.083}$ | $2.705^{\pm.06}$ |
| | **MotionGPT3**† | $0.533^{\pm.002}$ | $0.731^{\pm.002}$ | $0.826^{\pm.003}$ | $0.239^{\pm.008}$ | $2.797^{\pm.007}$ | $9.688^{\pm.107}$ | $1.560^{\pm.052}$ |
| Gen. & Und. | TM2T Guo et al. (2022c) | $0.424^{\pm.003}$ | $0.618^{\pm.003}$ | $0.729^{\pm.002}$ | $1.501^{\pm.046}$ | $3.467^{\pm.008}$ | $8.589^{\pm.058}$ | $\textbf{2.424}^{\pm.093}$ |
| | MotionGPT Jiang et al. (2023) | $0.492^{\pm.003}$ | $0.681^{\pm.003}$ | $0.733^{\pm.006}$ | $0.232^{\pm.008}$ | $3.096^{\pm.008}$ | $\underline{9.528}^{\pm.071}$ | $2.008^{\pm.084}$ |
| | MotionGPT-2 Wang et al. (2024) | $0.496^{\pm.002}$ | $0.691^{\pm.003}$ | $0.782^{\pm.004}$ | $0.191^{\pm.004}$ | $3.08^{\pm.013}$ | $9.571^{\pm.069}$ | $2.137^{\pm.022}$ |
| | LaMP Li et al. (2024b) | $\textbf{0.557}^{\pm.003}$ | $\textbf{0.751}^{\pm.002}$ | $\textbf{0.843}^{\pm.001}$ | $\underline{0.032}^{\pm.002}$ | $\underline{2.759}^{\pm.007}$ | $9.571^{\pm.069}$ | - |
| | MoTe Wu et al. (2024c) | $0.548^{\pm.002}$ | $0.737^{\pm.002}$ | $0.825^{\pm.002}$ | $0.075^{\pm.004}$ | $2.867^{\pm.012}$ | - | $2.399^{\pm.075}$ |
| | MG-MotionLLM Wu et al. (2025) | $0.516^{\pm.002}$ | $0.706^{\pm.002}$ | $0.802^{\pm.003}$ | $0.303^{\pm.010}$ | $2.952^{\pm.009}$ | $9.960^{\pm.073}$ | $2.125^{\pm.159}$ |
| | **MotionGPT3** | $\underline{0.553}^{\pm.003}$ | $\underline{0.747}^{\pm.002}$ | $\underline{0.837}^{\pm.003}$ | $0.208^{\pm.006}$ | $\textbf{2.725}^{\pm.008}$ | $9.700^{\pm.096}$ | $1.018^{\pm.038}$ |

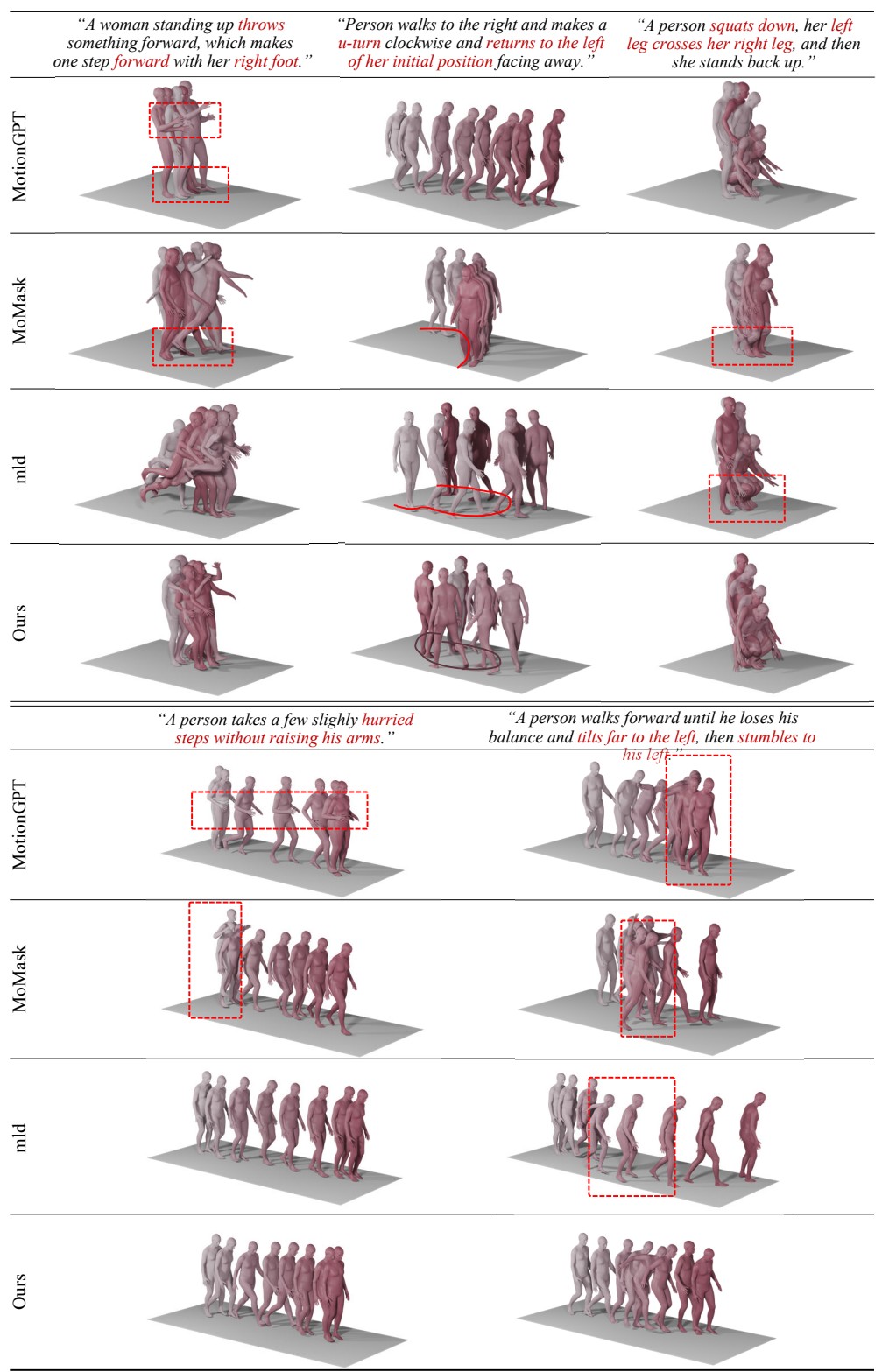

Figure 5: Qualitative comparison of text-driven motion generation on HumanML3D (Guo et al., 2022a). Baselines ( (Jiang et al., 2023; Guo et al., 2024; Xin et al., 2023)) are run with their official released checkpoints. Red annotations (text, boxes, curves) highlight prompt–motion mismatches. Our bimodal motion-language framework yields motions that with closer correspondence to the textual prompt and smoother temporal coherence.

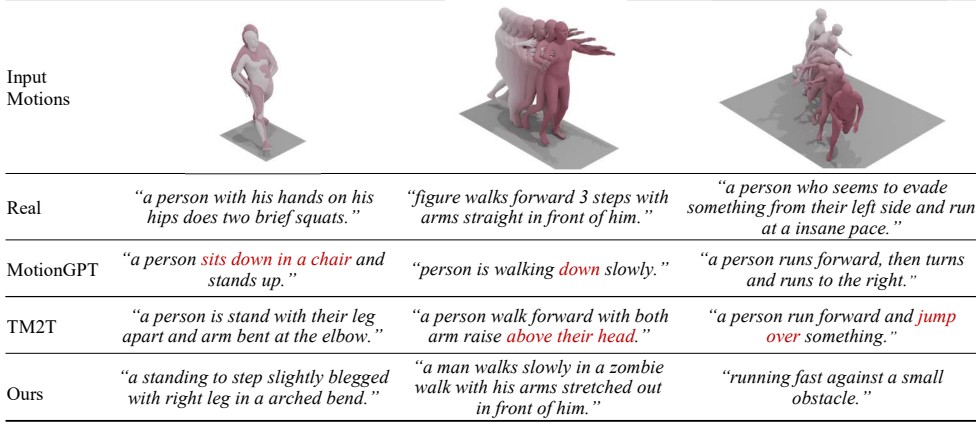

| | | | |
|---|---|---|---|
| Input Motions | | | |
| Real | *"a person with his hands on his hips does two brief squats."* | *"figure walks forward 3 steps with arms straight in front of him."* | *"a person who seems to evade something from their left side and run at a insane pace."* |
| MotionGPT | *"a person sits down in a chair and stands up."* | *"person is walking down slowly."* | *"a person runs forward, then turns and runs to the right."* |
| TM2T | *"a person is stand with their leg apart and arm bent at the elbow."* | *"a person walk forward with both arm raise above their head."* | *"a person run forward and jump over something."* |
| Ours | *"a standing to step slightly blegged with right leg in a arched bend."* | *"a man walks slowly in a zombie walk with his arms stretched out in front of him."* | *"running fast against a small obstacle."* |

Figure 6: Example results on motion caption. The misalignment is hilighted with red.

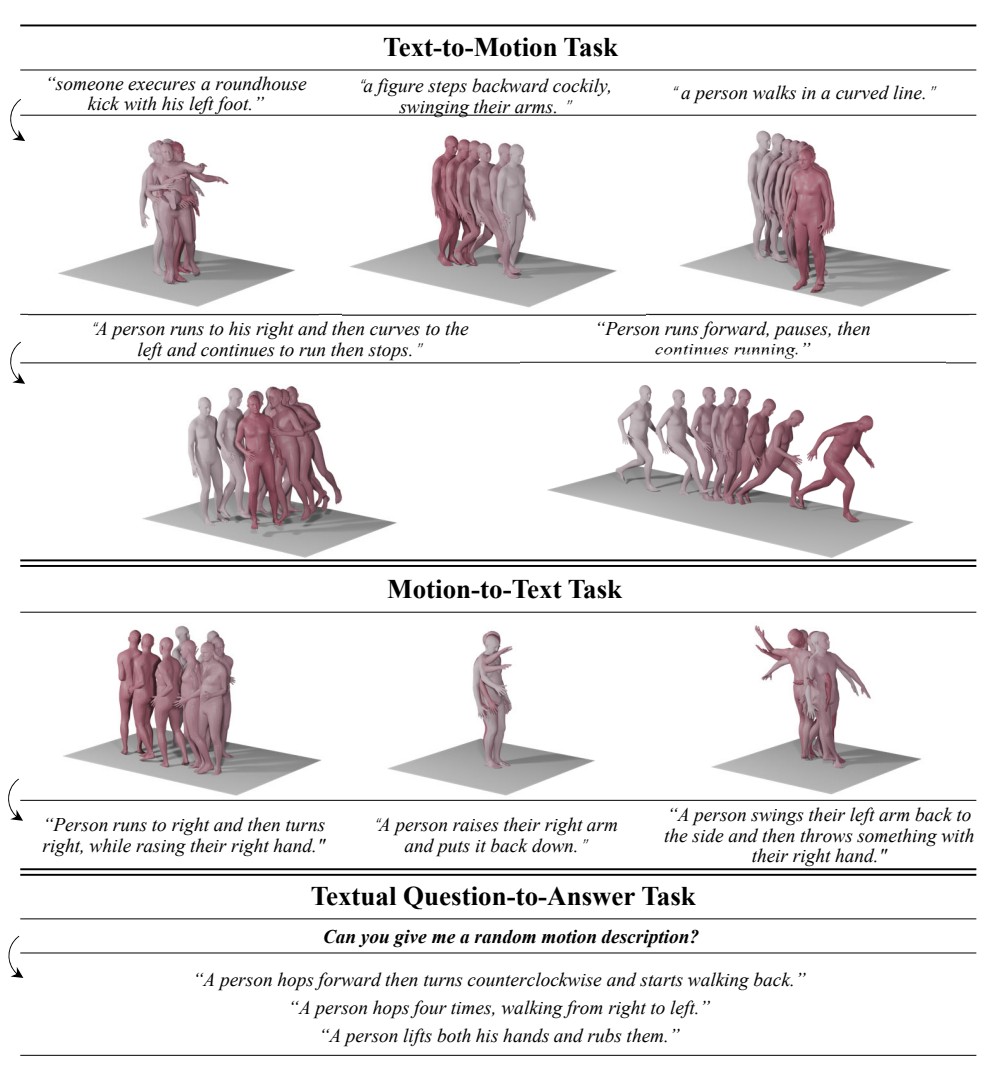

**Text-to-Motion Task**

*"someone execures a roundhouse kick with his left foot."*   *"a figure steps backward cockily, swinging their arms."*   *"a person walks in a curved line."*

*"A person runs to his right and then curves to the left and continues to run then stops."*   *"Person runs forward, pauses, then continues running."*

**Motion-to-Text Task**

*"Person runs to right and then turns right, while rasing their right hand."*   *"A person raises their right arm and puts it back down."*   *"A person swings their left arm back to the side and then throws something with their right hand."*

**Textual Question-to-Answer Task**

*Can you give me a random motion description?*

*"A person hops forward then turns counterclockwise and starts walking back."*
*"A person hops four times, walking from right to left."*
*"A person lifts both his hands and rubs them."*

Figure 7: Gallery for the results of MotionGPT3. Top: text-to-motion generation. Middle: motion-to-text captioning. Bottom: textual question answering about motion. Examples are produced by our unified model trained with instruction-based objectives (three-stage scheme). Animated visualizations are provided in the supplementary video.

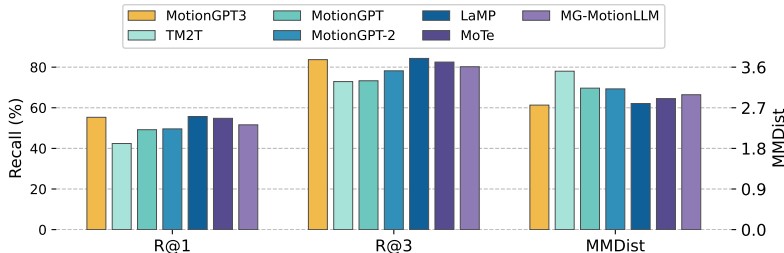

Figure 8: Comparison on text-to-motion, with recent approaches trained with unified tasks(Gen. & Und.). Our model performs betther than recent unified models: MotionGPT (Jiang et al., 2023), TM2T (Guo et al., 2022c), MotionGPT-2 (Wang et al., 2024), MG-MotionLLM (Wu et al., 2025), MoTe (Wu et al., 2024c), and comparable with LaMP (Li et al., 2024b).

Generally speaking, high evaluation scores indicates that the generated motions can correspond well with the text, and approximate high-fidelity motions. **However**, considering that the GT embeddings effectively represent an upper bound for matching scores, the practical significance of differences among methods achieving near or above GT performance might be limited. This reflects a limitation of the T2M evaluator, where the metrics are computed in a learned embedding space which relies on contrastive learning on HumanML3D. Methods that overfit to that space can saturate the proxy and even surpass the ground-truth reference, without a proportionate improvement in motion fidelity. Hence, differences among methods near or above the Real line are hard to interpret.

## C.2 EVALUATION WITH TMR

For a more nuanced assessment, we further evaluate with the TMR retrieval framework (Petrovich et al., 2023), under four gallery protocols: **All**, entire test set, **All with threshold**, gallery items whose textual similarity to the query exceeds a fixed threshold, **Dissimilar**, a 100-pair subset with mutually distant texts, and **Small batches**, mini-batch of size 32 to mimic Guo et al. (2022b) setting. We report text–motion retrieval with R@1/2/3/5/10 and MedR, results are summarized in Tab. 7.

Not all methods in Tab. 6 release TMR results or checkpoints, so the comparison is limited to publicly available models. Within this scope, although inferior to Li et al. (2024b) on T2M metrics (Fig. 8), our method achieves significantly stronger retrieval under the TMR evaluator and improves substantially over the T2M baseline Guo et al. (2022b) across protocols. These findings suggest that our model achieves more robust cross-modal alignment rather than overfitting to a specific evaluator.

Table 7: Retrieval on HumanML3D with the TMR evaluator (Petrovich et al., 2023). We report R@1/2/3/5/10 and MedR for text-motion retrieval under the four official protocols: (a) All, (b) All with threshold, (c) Dissimilar subset, and (d) Small batches (see Sec. C.2 for definitions). Results for TEMOS (Petrovich et al., 2022), T2M (Guo et al., 2022b), and TMR (Petrovich et al., 2023) are taken from the TMR paper. LaMP (Li et al., 2024b) is reported only for (d). MotionGPT and MotionGPT3 are evaluated with the released checkpoints using the official TMR code. MotionGPT3 attains strong performance across protocols.

| Protocol | Methods | Text-motion retrieval | | | | | | Motion-text retrieval | | | | | |
|---|---|---|---|---|---|---|---|---|---|---|---|---|---|
| | | R@1↑ | R@2↑ | R@3↑ | R@5↑ | R@10↑ | MedR↓ | R@1↑ | R@2↑ | R@3↑ | R@5↑ | R@10↑ | MedR↓ |
| All | TEMOS | 2.12 | 4.09 | 5.87 | 8.26 | 13.52 | 173.00 | 3.86 | 4.54 | 6.94 | 9.38 | 14.00 | 183.25 |
| | T2M | 1.80 | 3.42 | 4.79 | 7.12 | 12.47 | 81.00 | 2.92 | 3.74 | 6.00 | 8.36 | 12.95 | 81.50 |
| | TMR | 5.68 | 10.59 | 14.04 | 20.34 | 30.94 | 28.00 | 9.95 | 12.44 | 17.95 | 23.56 | 32.69 | 28.50 |
| | MotionGPT | 7.16 | 12.50 | 15.85 | 21.53 | 30.20 | 38.00 | 11.31 | 13.91 | 19.39 | 24.13 | 31.80 | 36.25 |
| | MotionGPT3 | **9.60** | **17.36** | **22.45** | **30.43** | **41.06** | **17.00** | **14.90** | **18.20** | **24.43** | **31.32** | **40.72** | **17.50** |
| All with threshold | TEMOS | 5.21 | 8.22 | 11.14 | 15.09 | 22.12 | 79.00 | 5.48 | 6.19 | 9.00 | 12.01 | 17.10 | 129.00 |
| | T2M | 5.30 | 7.83 | 10.75 | 14.59 | 22.51 | 54.00 | 4.95 | 5.68 | 8.93 | 11.64 | 16.94 | 69.50 |
| | TMR | 11.60 | 15.39 | 20.50 | 27.72 | 38.52 | 19.00 | 13.20 | 15.73 | 22.03 | 27.65 | 37.63 | 21.50 |
| | MotionGPT | 14.32 | 21.01 | 25.94 | 33.39 | 43.84 | 15.00 | 14.42 | 16.83 | 22.70 | 27.69 | 35.06 | 30.50 |
| | MotionGPT3 | **20.73** | **27.03** | **34.03** | **42.66** | **52.97** | **9.00** | **19.34** | **22.40** | **29.40** | **36.91** | **46.30** | **13.00** |
| Dissimilar subset | TEMOS | 33.00 | 42.00 | 49.00 | 57.00 | 66.00 | 4.00 | 35.00 | 44.00 | 50.00 | 56.00 | 70.00 | 3.50 |
| | T2M | 34.00 | 48.00 | 57.00 | 72.00 | 84.00 | 3.00 | 34.00 | 47.00 | 59.00 | 72.00 | 83.00 | 3.00 |
| | TMR | 47.00 | 61.00 | 71.00 | 80.00 | 86.00 | 2.00 | 48.00 | 63.00 | 69.00 | 80.00 | 84.00 | 2.00 |
| | MotionGPT | 51.00 | 64.00 | 71.00 | 74.00 | 80.00 | 1.00 | 53.00 | 62.00 | 68.00 | 76.00 | 81.00 | 1.00 |
| | MotionGPT3 | **68.00** | **77.00** | **85.00** | **92.00** | **95.00** | **1.00** | **63.00** | **73.00** | **83.00** | **89.00** | **93.00** | **1.00** |
| Small batches | TEMOS | 40.49 | 53.52 | 61.14 | 70.96 | 84.15 | 2.33 | 39.96 | 53.49 | 61.79 | 72.40 | 85.89 | 2.33 |
| | T2M | 52.48 | 71.05 | 80.65 | 89.66 | 96.58 | 1.39 | 52.00 | 71.21 | 81.11 | 89.87 | 96.78 | 1.38 |
| | TMR | 67.16 | 81.32 | 86.81 | 91.43 | 95.36 | 1.04 | 67.97 | 81.20 | 86.35 | 91.70 | 95.27 | 1.03 |
| | LaMP | 67.18 | 81.90 | 87.04 | 92.00 | 95.73 | - | 68.02 | 82.10 | 87.50 | 92.20 | 96.90 | - |
| | MotionGPT | 58.07 | 69.91 | 74.34 | 79.17 | 86.36 | 1.18 | 58.71 | 69.64 | 74.36 | 79.45 | 86.02 | 1.16 |
| | MotionGPT3 | **74.25** | **86.70** | **91.29** | **94.82** | **97.35** | **1.00** | **74.00** | **86.86** | **91.04** | **94.62** | **97.35** | **1.00** |

## C.3 MOTION BRANCH WITH CROSS-MODAL CONNECTION

Our hybrid model allows asymmetric capacities for the text and motion branches and supports different patterns of cross-modal information exchange. In this section we focus on *where* to place cross-modal attention (CMA) in the backbone for the text-to-motion task, keeping all other factors fixed (Tab. 9). Ablation on branch capacity is deferred to Sec. C.4.

We explore several CMA schedules that differ only in the layers where cross-modal connections are enabled(Tab. 9). Across paired settings with the same spacing pattern, shifting the CMA blocks to later layers typically improves generation quality and distribution similarity to the ground-truth (i.e., lower FID and MMDist, higher R-Precision). This is most evident in $B_1$ v.s. $B_2$: both use uniformly spaced CMA with identical count, $B_1$ enable CMA from the first to the second-last layer, while $B_2$ shifts them by one layer to span the second through the last layer. Despite the minor offset, $B_2$ achieves noticeably better scores. The same tendency appears in $B3$–$B4$, $C1$–$C3$, and $D_1$-$D_2$, where the distribution pattern is matched but the CMA positions differ.

Table 8: Cross-modal attention(CMA) configurations used in ablation. (a) Layer-wise CMA schedules for configurations A–D across the 12-layer backbones. Within each branch, the symbol $\leftrightarrows$ marks a cross-modal attention (CMA) operation at that layer, blanks empty indicate intra-modal attention only. (b) Schematics diagrams for configurations $A$, $B_1$, $C_1$.

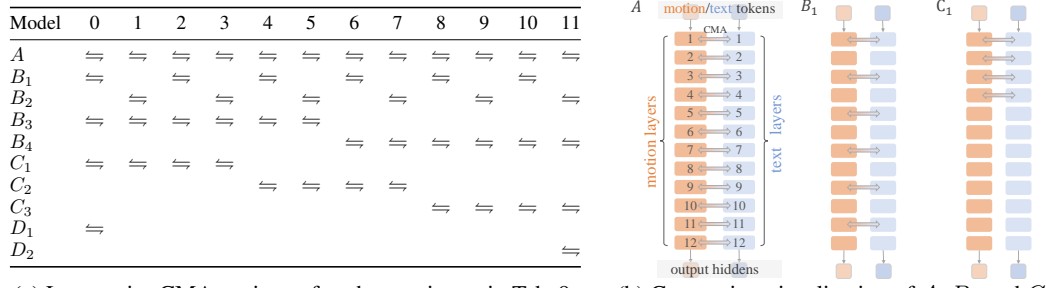

| Model | 0 | 1 | 2 | 3 | 4 | 5 | 6 | 7 | 8 | 9 | 10 | 11 |
|---|---|---|---|---|---|---|---|---|---|---|---|---|
| $A$ | $\leftrightarrows$ | $\leftrightarrows$ | $\leftrightarrows$ | $\leftrightarrows$ | $\leftrightarrows$ | $\leftrightarrows$ | $\leftrightarrows$ | $\leftrightarrows$ | $\leftrightarrows$ | $\leftrightarrows$ | $\leftrightarrows$ | $\leftrightarrows$ |
| $B_1$ | $\leftrightarrows$ | | $\leftrightarrows$ | | $\leftrightarrows$ | | $\leftrightarrows$ | | $\leftrightarrows$ | | $\leftrightarrows$ | |
| $B_2$ | | $\leftrightarrows$ | | $\leftrightarrows$ | | $\leftrightarrows$ | | $\leftrightarrows$ | | $\leftrightarrows$ | | $\leftrightarrows$ |
| $B_3$ | $\leftrightarrows$ | $\leftrightarrows$ | $\leftrightarrows$ | $\leftrightarrows$ | $\leftrightarrows$ | $\leftrightarrows$ | | | | | | |
| $B_4$ | | | | | | | $\leftrightarrows$ | $\leftrightarrows$ | $\leftrightarrows$ | $\leftrightarrows$ | $\leftrightarrows$ | $\leftrightarrows$ |
| $C_1$ | $\leftrightarrows$ | $\leftrightarrows$ | $\leftrightarrows$ | $\leftrightarrows$ | | | | | | | | |
| $C_2$ | | | | | $\leftrightarrows$ | $\leftrightarrows$ | $\leftrightarrows$ | $\leftrightarrows$ | | | | |
| $C_3$ | | | | | | | | | $\leftrightarrows$ | $\leftrightarrows$ | $\leftrightarrows$ | $\leftrightarrows$ |
| $D_1$ | $\leftrightarrows$ | | | | | | | | | | | |
| $D_2$ | | | | | | | | | | | | $\leftrightarrows$ |

(a) Layer-wise CMA settings of each experiment in Tab. 9.    (b) Connection visualization of $A$, $B_1$ and $C_1$.

Table 9: Quantitative results for several CMA settings on T2M with 200K training iterations, settings visualized in Tab. 8. The text branch is pretrained GPT-2 (124M) and the motion branch has 114M parameters. Increasing the number of CMA layers and placing them later in the network generally improves performance, and A is the best among tested settings. See Sec. C.3 for further analysis.

| | R@1↑ | R@2↑ | R@3↑ | FID↓ | MMDist↓ | DIV→ | MModality↑ |
|---|---|---|---|---|---|---|---|
| Real | 0.518 | 0.713 | 0.813 | - | 2.811 | 9.976 | - |
| $A$ | **0.536** | **0.728** | **0.819** | 0.241 | **2.767** | 10.379 | 2.454 |
| $B_1$ | 0.502 | 0.707 | 0.807 | 0.311 | 2.895 | 10.318 | 2.489 |
| $B_2$ | 0.508 | 0.714 | 0.812 | 0.288 | 2.8637 | 10.261 | 2.315 |
| $B_3$ | 0.508 | 0.712 | 0.811 | 0.22 | 2.879 | 10.405 | 2.664 |
| $B_4$ | 0.514 | 0.716 | 0.814 | 0.243 | 2.839 | 10.347 | 2.534 |
| $C_1$ | 0.506 | 0.701 | 0.801 | 0.236 | 2.894 | 10.386 | 2.684 |
| $C_2$ | 0.502 | 0.702 | 0.795 | 0.285 | 2.948 | 10.333 | 2.631 |
| $C_3$ | 0.503 | 0.705 | 0.803 | 0.171 | 2.886 | 10.221 | 2.819 |
| $D_1$ | 0.473 | 0.663 | 0.767 | 0.283 | 3.105 | **10.176** | **3.770** |
| $D_2$ | 0.477 | 0.672 | 0.777 | **0.164** | 3.092 | 10.189 | 3.197 |

## C.4 ABLATION ON MODEL SIZE

We examine model size along three axes while keeping all other settings fixed: (i) the overall capacity, achieved by scaling the text and motion branches proportionally, (ii) motion-branch capacity with a fixed text branch, and (iii) language-backbone size with a comparable motion branch.

Tab. 10 compares overall **backbone sizes**. With roughly 3× parameters, the medium model yields modest gains on R@k and MMDist despite slightly higher FID. This suggests greater capacity helps capture high-level semantics, though realizing its full benefit may require careful optimization.

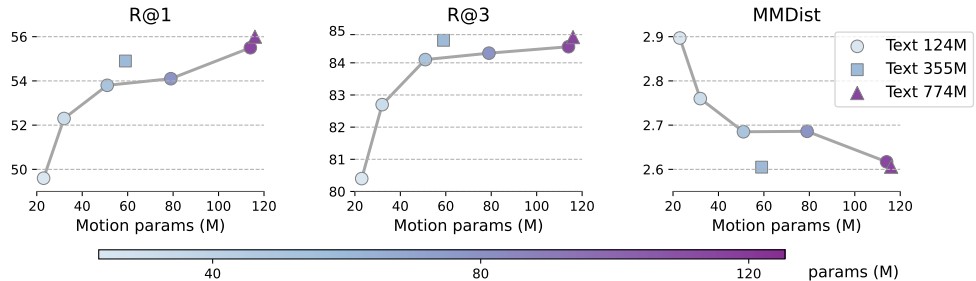

Figure 9: Ablation on branch capacity on motion generation. All models are trained for 200K iterations. A 124M text branch already performs competitively with much larger backbones (355M/774M). Our model can achieve competitive performance with only halfed motion parameters (∼51M).

With the text branch fixed to GPT-2 small (124M), we **scale the motion branch** from 23M to 114M parameters by setting hidden size to 76, 192, 384, 576, 768 (Fig. 9). Increasing motion capacity generally improves text–motion alignment (higher R@k) and reduces MMDist, while diversity remaining stable. in a high level A half-sized motion branch (∼51M) already offers a strong trade-off, delivering competitive overall performance, and, the best FID among the 124M-text configurations.

**Text-branch Size.** To isolate the effect of the language backbone, we replace GPT-2 small (124M) with GPT-2 medium (355M) and GPT-2 large (774M), keeping the motion branch of comparable size (pairs: 355M/59M vs. 124M/51M, and 774M/116M vs. 124M/114M). Larger text branches further improve alignment and lower MMDist, and tend to increase diversity/MModality.

Table 10: Effect of GPT-2 backbone size and CFG guidance scale $\omega$ on text-to-motion task. All models are trained for 200K iterations. The medium backbone, with more parameters (692M v.s. 238M) consistently outperforms the small model.

| Size | Text Params | Total Params | $\omega$ | R@1↑ | R@2↑ | R@3↑ | FID↓ | MMDist↓ | DIV→ |
|------|-------------|--------------|----------|------|------|------|------|---------|------|
| small | 124M | 238M | 1.0 | 0.534 | 0.739 | 0.842 | 0.222 | 2.61 | 10.256 |
| medium | 355M | 692M | 1.0 | 0.558 | 0.756 | 0.852 | 0.235 | 2.553 | 10.238 |
| small | 124M | 238M | 3.0 | 0.552 | 0.759 | 0.852 | 0.173 | 2.554 | 10.239 |
| medium | 355M | 692M | 3.0 | 0.568 | 0.766 | 0.860 | 0.192 | 2.489 | 10.084 |

## C.5 VAE AND DIFFUSION HEAD

**Ablation on Diffusion Head.** We ablate the diffusion head $\mathcal{H}$ in our motion branch to study how conditioning design affects generation. We vary (i) supervision with a diffusion head $\mathcal{H}$ v.s. direct MSE regression, (ii) the mapping from backbone hidden states to the diffusion condition (multi-head attention, MHA vs. linear layer), (iii) the number of motion holders $h\_num \in \{1, 4, 8\}$ used to query hidden states from the autoregressive backbone, and (iv) classifier-free guidance (CFG) at sampling. All variants are trained for 200K iterations with results in Tab. 11.

Table 11: Ablation of motion generation head and loss on HumanML3D. All variants are trained on T2M task for 200k iterations, with same backbone and data. We vary: (i) supervision with diffusion head $\mathcal{H}$ (Diff.) or direct MSE regression (MSE), (ii) mapping of multi-head attention (MHA) or Linear layer in $\mathcal{H}$, (iii) number of motion holder <motion_out> ($h\_num$) (i.e., the count of hidden states passed from the backbone to $\mathcal{H}$), and (iv) classifier-free guidance (CFG) at sampling.

| ID | Loss | $h\_num$ | Head | CFG | R@1↑ | R@2↑ | R@3↑ | FID↓ | MMDist↓ | DIV→ | MModality↑ |
|----|------|----------|------|-----|------|------|------|------|---------|------|------------|
| (a) | Diff. | MHA | 4 | ✓ | **0.547** | **0.751** | **0.850** | **0.149** | **2.578** | 10.041 | 2.265 |
| (b) | Diff. | MHA | 8 | ✗ | 0.531 | 0.733 | 0.836 | 0.185 | 2.655 | 10.154 | 2.198 |
| (c) | Diff. | MHA | 4 | ✗ | 0.529 | 0.730 | 0.839 | 0.166 | 2.645 | 10.012 | 2.350 |
| (d) | Diff. | MHA | 1 | ✗ | 0.525 | 0.729 | 0.831 | 0.164 | 2.678 | 10.090 | 2.514 |
| (e) | Diff. | Linear | 4 | ✗ | 0.521 | 0.731 | 0.829 | 0.178 | 2.713 | 9.985 | 2.603 |
| (f) | Diff. | Linear | 1 | ✗ | 0.525 | 0.729 | 0.829 | 0.283 | 2.689 | 10.069 | **2.719** |
| (g) | MSE | Linear | 4 | - | 0.518 | 0.725 | 0.823 | 0.276 | 2.705 | 9.758 | 2.175 |

Following MotionGPT's evaluation protocol (Jiang et al., 2023), results are averaged over two runs.
Best results are **bold**, second-best are underlined. The default configuration is gray-shaded.

We observe that (i) Direct MSE supervision (the last row) yields the weakest performance, confirming the benefit of diffusion-based training. (ii) Increasing $h\_num$ (b-d) enriches the conditioning signal and improves retrieval accuracy R@k, but also raises FID, suggesting a harder denoising problem. A moderate setting of $h\_num = 4$ offers the best trade-off. (iii) With $h\_num = 4$ and no CFG (c)(e), an MHA head outperforms a linear mapper, achieving higher R@3 (0.529 v.s. 0.521), lower MMDist ( 2.645 v.s.2.713), and lower FID (0.166 v.s. 0.178) The advantage is even larger under sparse conditioning ($h\_num = 1$ in (d)(e), FID: 0.164 vs. 0.283); (iv) Enabling CFG on the same configuration further improves both alignment and fidelity. Accordingly, we adopt diffusion supervision with an MHA head and $h\_num = 4$ as the default.

**Guidance scale.** We sweep the CFG guidance scale $\omega$ on the unified model (results in Tab. 7). Note that guidance is applied only to *motion generation*. Moderate guidance performs best: $\omega = 5.0$ minimizes FID on the T2M, while $\omega = 3.0$ yeilds the best R-Precision and MultiModal Distance. Very weak ($\omega = 1$) or overly strong ($\omega \geq 10$) guidance degrades alignment and diversity. We thus use $\omega = 4$ in all main results.

Table 12: Ablation on guidance scale $\omega$ in CFG, for motion latent diffusion on HumanML3D, with model trained on unified tasks. Best and second-best results are **bold** and underlined.

| $\omega$ | R@1 | R@2 | R@3 | FID↓ | MMDist ↓ | DIV→ | MModality↑ |
|---|---|---|---|---|---|---|---|
| Real | 0.519 | 0.724 | 0.820 | 0.002 | 2.753 | 9.941 | - |
| 1 | 0.534 | 0.727 | 0.828 | 0.143 | 2.714 | 10.086 | **1.717** |
| 2 | **0.555** | 0.754 | 0.843 | 0.123 | 2.601 | 10.006 | 1.321 |
| 3 | 0.554 | **0.756** | **0.850** | 0.103 | **2.585** | 9.926 | 1.272 |
| 4 | 0.552 | 0.753 | 0.848 | 0.098 | 2.589 | 9.911 | 1.258 |
| 5 | 0.552 | 0.752 | 0.848 | **0.094** | 2.593 | 9.906 | 1.248 |
| 6 | 0.546 | 0.751 | 0.849 | **0.094** | 2.598 | 9.900 | 1.243 |
| 10 | 0.546 | 0.748 | 0.844 | 0.109 | 2.620 | 9.870 | 1.281 |
| 15 | 0.541 | 0.739 | 0.839 | 0.12 | 2.653 | 9.873 | 1.312 |
| 20 | 0.533 | 0.728 | 0.826 | 0.134 | 2.739 | 9.827 | 1.385 |

Following MotionGPT's protocol, results are averaged over two runs. The default configuration is gray-shaded.

## C.6 EFFECTIVENESS OF TRAINING SCHEME

We adopt a three-stage schedule (see Sec. 3.4, Fig. 2): SI, text-to-motion (T2M) pretraining; SII, cross-modal alignment with joint optimization on T2M and motion-to-text (M2T) (SII); and SIII, joint fine-tuning. We evaluate three settings: (i) **Three Stages**, the full three-stage schedule, (ii) **Two Stage**, a two-stage schedule without SI, and (iii) **Trained Text Branch**, a two-stage variant in which the text branch is unfrozen during SI–SII, rendering SIII unnecessary. We report results on both generation (T2M) and understanding (M2T) in Tab. 13.

Table 13: Training-scheme evaluation on HumanML3D (Guo et al., 2022a), with protocol in Jiang et al. (2023). Stage1: T2M Pre-training, Stage2: Cross-Modal Alignment, Stage3: Joint Fintuning. ✓marks enabled stages, while colors encode the state of text branch, updated or frozen. Jointly updating the text branch from the start improves early T2M in SI but degrades final T2M after SII and markedly lowers M2T scores (rows "Trained Text Branch").

| Type | Stage1 | Stage2 | Stage3 | Text-to-Motion | | | Motion-to-Text | | |
|---|---|---|---|---|---|---|---|---|---|
| | | | | R@3 ↑ | FID ↓ | MMDist ↓ | R@1↑ | Bleu@4↑ | BertScore↑ |
| Three Stages | ✓ | ✗ | ✗ | 0.826 | 0.239 | 2.797 | - | - | - |
| | ✓ | ✓ | ✗ | 0.831 | 0.215 | 2.755 | 0.571 | 18.328 | 33.993 |
| | ✓ | ✓ | ✓ | **0.837** | **0.208** | **2.725** | 0.573 | 19.412 | 35.231 |
| Two Stages | ✗ | ✓ | ✗ | 0.755 | 0.298 | 3.213 | 0.561 | 18.295 | 34.676 |
| | ✗ | ✓ | ✓ | 0.772 | 0.325 | 3.108 | 0.573 | 18.277 | 35.546 |
| Trained Text Branch | ✓ | ✗ | - | 0.822 | 0.239 | 2.832 | - | - | - |
| | ✓ | ✓ | - | 0.801 | 0.243 | 2.942 | 0.505 | 14.119 | 33.385 |

Following MotionGPT's protocol, metrics are averaged over two runs.
Best and second-best results are **bold** and underlined.

**Text-to-motion Pre-training** and **Cross-Modal Alignment**. Pretraining on T2M (SI) yields strong motion generation and provides a motion-specialized initialization. Entering SII confers M2T capability and further improves T2M (alignment improves and MMDist/FID drop), indicating that explicit cross-modal alignment benefits both directions. Training directly with multi-task objectives from scratch (i.e., without SI) markedly degrades T2M quality, even after subsequent joint optimization, underscoring the importance of a motion-specific warm start, i.e., initializing the motion branch by T2M pretraining.

**Joint Fine-tuning.** Once SII has established cross-modal alignment, SIII yields modest gains, primarily stabilizing M2T while preserving T2M, and thus serves as a light refinement. The "Two Stages" variants (SII+SIII without SI) show that joint optimization can boost both tasks when the model is under-initialized. However, it also degrades the language branch's competence, leading to worse T2M and M2T than the full three-stage schedule. Although incremental gains beyond SI+SII are modest, SIII can mitigate residual negative transfer and calibrate cross-modal alignment under noisy or shifted training conditions, yielding more stable results.

**Freezing v.s. training the text branch.** To promote modality-specific representations and reduce negative transfer onto a well-trained language branch, we propose to freeze the text branch in SI–SII. Freezing preserves linguistic competence while the motion branch specializes. By contrast, updating all parameters from the start ("Trained Text Branch") can give slightly higher T2M scores in SI (e.g., R@3 0.834 vs. 0.820; MMDist 2.698 vs. 2.787), but after SII these models exhibit degraded T2M and notably weaker M2T (e.g., BertScore 0.713; Bleu@4 3.577), consistent with 'catastrophic forgetting'. We attribute this decline to negative transfer from the new motion branch onto the text branch during early training. In practice, keeping the LM frozen lets the motion branch learn a more stable, modality-specific space and achieve more reliable alignment under *limited* paired data.

In summary, SI provides essential motion-specific initialization; SII delivers the bulk of cross-modal gains; SIII offers small, stabilizing improvements. Freezing the text branch through SI–SII prevents loss of linguistic ability and yields the best overall balance between understanding and generation.

## C.7 CHOICE OF BACKBONE

We instantiate our framework with three language backbones of GPT-2 Radford et al. (2019), Flan-T5-base Chung et al. (2022), and Qwen2.5-0.5B-Instruct Qwen et al. (2025), under both single- and dual-stream architectures. In the dual-stream setting, we set the hidden dimension of the motion branch to 374 for T5 and 448 for Qwen2.5, so as to roughly balance the number of newly introduced parameters. In preliminary experiments, using a motion branch with the same size as the text branch in Qwen2.5 led to unstable training and degraded performance, presumably because a large number of parameters trained from scratch would require a much larger benchmark. Exploring larger-capacity motion branches with stronger pretraining or larger-scale datasets is left for future work.

As shown in Tab. 14, across all three language backbones, introducing our dual-stream architecture consistently improves retrieval metrics and MMDist over the single-stream variants, indicating that our framework is not tied to a particular language model and scales well to more recent LLMs.

Table 14: Text-to-motion generation results with GPT-2, Flan-T5, and Qwen2.5 backbones under single- and dual-stream architectures. All models are trained on 2 NVIDIA RTX 3090 GPUs, for 400 epochs in single-stream setting and 200 epochs in dual-streamm setting. For each configuration, we report metrics from the checkpoint with the lowest validation FID.

| Architecture | Backbone | Parameters | R@1 | R@2 | R@3 | FID↓ | MMDist↓ | Diversity→ |
|---|---|---|---|---|---|---|---|---|
| Real | - | - | 0.511 | 0.703 | 0.797 | 0.002 | 2.974 | 9.503 |
| single | GPT2 | 152M | 0.513 | 0.708 | 0.808 | 0.485 | 2.962 | 9.633 |
| | T5 | 275M | 0.491 | 0.681 | 0.777 | 0.243 | 3.102 | 9.365 |
| | Qwen | 524M | 0.483 | 0.669 | 0.768 | 0.491 | 3.143 | 9.653 |
| dual | GPT2 | 238M | 0.533 | 0.731 | 0.826 | 0.239 | 2.797 | 9.688 |
| | T5 | 369M | 0.509 | 0.702 | 0.799 | 0.285 | 2.978 | 9.772 |
| | Qwen | 617M | 0.542 | 0.739 | 0.832 | 0.309 | 2.782 | 9.878 |

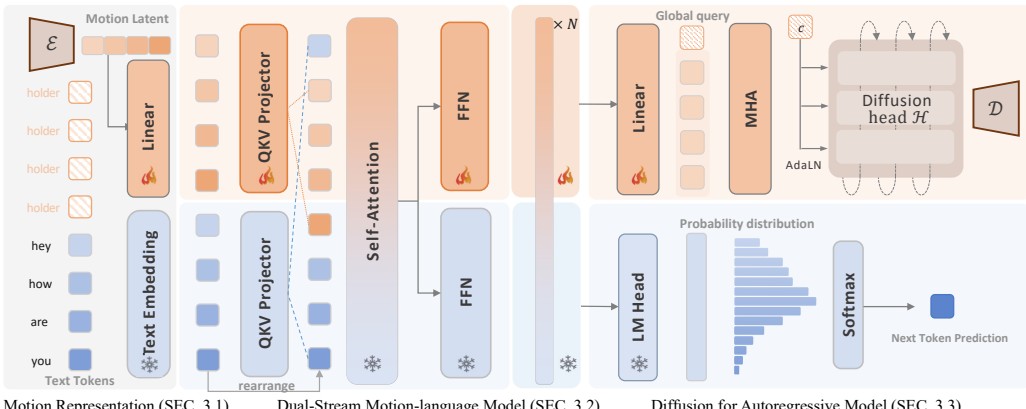

Figure 10: Details of our bimodal motion-language model. Motion and text inputs are encoded by separate branches and then reordered to their original sequence order before cross-modal self-attention. After $N$ hybrid layers, text is generated autoregressively by next-token prediction, while motion is produced via a diffusion head $\mathcal{H}$. Panels correspond to motion representation (Sec. 3.1), the dual-stream motion–language backbone (Sec. 3.2), and the diffusion module (Sec. 3.3).

# D    DETAILS ON MOTIONGPT3

## D.1    MOTION GENERATION IN UNIFIED MODEL

**Diffusion Loss** We use a diffusion head $\mathcal{H}$ to map backbone hidden states into a denoised motion latent. As illustrated in Figs. 2 and 10, we insert $K$ *motion holder* tokens <motion_out> as queries to extract the corresponding output states from the motion branch. Let the backbone hidden size be $d_t$ and the motion latent size be $d_m$ (we set $d_t{=}768$, $d_m{=}256$ in our default model). After one forward pass, the queried states form $h \in \mathbb{R}^{K \times d_t}$. An MHA pooling module aggregates $h$ and produces a global condition vector $c \in \mathbb{R}^{1 \times d_m}$ via an internal mapping to match the motion-latent dimensionality, which is then fused with the timestep embedding $\tau(t)$ to condition $c$. With the cumulative product of the noise schedule $\bar{\alpha}_t {=} \prod_{s=1}^{t} \alpha_s$, , we sample $t \sim \mathcal{U}\{1, \ldots, T\}$ and corrupt the ground-truth motion latent $z_0 \in \mathbb{R}^{d_m}$ by

$$z_t = \sqrt{\bar{\alpha}_t}\, z_0 + \sqrt{1 - \bar{\alpha}_t}\, \epsilon, \quad \epsilon \sim \mathcal{N}(0, I), \tag{2}$$

Given a noisy latent $z_t \in \mathbb{R}^{1 \times d_m}$ at timestep $t$, the head predicts the noise $\hat{\epsilon}_\theta$ and is trained with the standard $\epsilon$-prediction objective

$$L_{diff} \;=\; \mathcal{E}_{z_0, \epsilon, t} \| \epsilon - \hat{\epsilon}_\theta(\alpha_t z_0 + \sigma_t \epsilon, t, c) \|_2^2, \quad \hat{\epsilon}_\theta {=} \mathcal{H}(z_t, t, c) \tag{3}$$

where $\alpha_t$, $\sigma_t$ follow a linear schedule equivalent to the forward noising process. At inference, we start from $z_T \sim \mathcal{N}(0, I)$ iteratively denoise with the sampler of $\mathcal{H}$ down to $t{=}0$ to obtain $\hat{z}_0$, which is then decoded by the motion decoder $\mathcal{D}$ into a raw motion sequence.

**Architecture of $\mathcal{H}$.** The diffusion head first processes the $K$ queried hidden states with a TransformerEncoderLayer and aggregates them via multi-head attention pooling into a single condition vector $c$. We fuse $c$ with the timestep embedding and modulate each block via AdaLN. $\mathcal{H}$ consists of a stack of 1024-wide residual blocks. Each block applies AdaLN followed by a two-layer MLP with SiLU nonlinearity, and the final block projects to the noise prediction $\hat{\epsilon}_\theta$.

**Cross-Entropy Loss for Boundary Tokens** To delimit motion from text decoding, we introduce two boundary tokens<som> (start of motion) and <eom> (end of motion). At inference, once the LM predicts <som> via next-token prediction, we generate the motion latent in a single forward pass, with $K$ <motion_out> holders concatenated to the sequence, and then append <eom> deterministically. During training, we apply cross-entropy only to the <som> prediction, and <eom> is not supervised.

## D.2 MOTION VAE

**AutoEncoder** We adopt a Transformer-based motion VAE (Xin et al., 2023) with an encoder $\mathcal{E}$ and a decoder $\mathcal{D}$ that maps an $M$-frame motion sequence $m^{1:M}$ to a compact continuous latent $z \in \mathbb{R}^{n \times d}$ ($n = 1, d = 256$) and reconstructs the motion via $m^{1:M} = \mathcal{D}(z) = \mathcal{D}(\mathcal{E}(m^{1:M}))$. Both $\mathcal{E}$ and $\mathcal{D}$ are transformers (Vaswani et al., 2017) with long skip connection (Ronneberger et al., 2015), and without the action biases used in Petrovich et al. (2021). This design yields an expressive latent space that supports accurate semantic understanding and high-fidelity, diverse motion synthesis.

**Architecture** Given an input motion sequence $m^{1:M}$ of length $M$, the encoder $\mathcal{E}$ processes the sequence together with a small set of learnable distribution tokens and outputs the Gaussian parameters $(\mu_m, \sigma_m)$. A latent $z$ is sampled by reparameterization $z = \mu_m + \sigma_m \epsilon, \epsilon \, in \, \mathcal{N}(0, I)$. The decoder $\mathcal{D}_{mld}$ performs cross-attention over the latent vector $z$ to query $L$ motion tokens, which are then projected back into $\hat{m}^{1:L}$ in the raw motion space.

$$\hat{m}^{1:L} = \mathcal{D}(\mathcal{E}(m^{1:M}) \tag{4}$$

**Loss** We train the motion VAE with reconstruction term over framewise poses and KL regularizer on the latent, following standard practice in VAEs (Kullback & Leibler, 1951; Kingma & Welling, 2013). Let $m^{1:M}$ denote a sequence of $M$ ground-truth poses $m_t \in \mathbb{R}^{263}$ and $\hat{m}^{1:M}$ the decoder outputs. The encoder produces a Gaussian posterior $q_\phi(z \,|\, m^{1:M}) = \mathcal{N}(\mu, \mathrm{diag}(\sigma^2))$. The objective is

$$\mathcal{L} = \mathcal{L}_{\mathrm{rec}} + \lambda_{\mathrm{KL}} \, \mathcal{L}_{\mathrm{KL}}, \tag{5}$$

with

$$\mathcal{L}_{\mathrm{rec}} = \frac{1}{T} \sum_{t=1}^{T} \lVert m_t - \hat{m}_t \rVert_2^2, \tag{6}$$

$$\mathcal{L}_{\mathrm{KL}} = D_{\mathrm{KL}}\big(\mathcal{N}(\mu, \mathrm{diag}(\sigma^2)) \,\big\|\, \mathcal{N}(0, I)\big). \tag{7}$$

For completeness, the KL term admits the closed form $\mathcal{L}_{\mathrm{KL}} = \frac{1}{2} \sum_j \big(\mu_j^2 + \sigma_j^2 - \log \sigma_j^2 - 1\big)$.

**Raw Motion Representation** Following Guo et al. (2022a), each frame $m^i \in \mathbb{R}^{263}$ concatenates a tuple of root angular velocity $\dot{r}^a \in \mathbb{R}$ along Y-axis, root linear velocities $(\dot{r}^x, \dot{r}^z) \in \mathbb{R}$ on XZ-plane, root height $r^y \in \mathbb{R}$, local joints positions $j^p \in \mathbb{R}^{3N_j}$, velocities $j^v \in \mathbb{R}^{3N_j}$, and rotations $j^r \in \mathbb{R}^{6N_j}$ in root space, with $N_j$ denotes the joint number, and binary foot ground contact features $c^f \in \mathbb{R}^4$ by thresholding the heel and toe joint velocities. This finally results in $m^i = \{\dot{r}^a, \dot{r}^x, \dot{r}^z, r^y, j^p, j^v, j^r, c^f\}$.

## D.3 METRICS DEFINITIONS

We adopt standard metrics for text–motion alignment, motion quality and diversity, caption quality, and (for VAE analysis) motion reconstruction. Unless noted, features are computed with the official HumanML3D/T2M evaluator (Guo et al., 2022b), with motion encoder $\phi(m)$ and text encoder $\psi(t)$.

**Text–motion alignment** To evaluate semantic consistency between generated motions and input texts, we adopt motion-text retrieval precision (R-Precision) at Top-k(R@k), and the Multimodal Distance (MMDist), which measures the embedding-space distance between paired modalities. **R@k** measures retrieval accuracy within a candidate set: for each query (text or motion), we rank candidates of the other modality by cosine similarity and report the fraction of cases where the paired item appears in the top-k. **MMDist** is the average embedding distance between paired items:

$$\mathrm{MMDist} = \frac{1}{N} \Sigma_{n=1}^{N} \lVert \phi(m_n) - \psi(t_n) \rVert_2 \tag{8}$$

**Motion quality** **FID** assess how closely generated motions match ground truth ones in feature space, indicating overall quality, and is computed between the Gaussian fits of $\{\phi(m)\}$ for generated and ground-truth motions in the evaluator feature space.

**Diversity**   Diversity (DIV) measures feature variation across samples, and MultiModality (MM), which quantifies variation among motion generations from the same textual description. Following Guo et al. (2022b); Xin et al. (2023), we randomly sample all generated motions into two subsets, $\{x_i\}_{i=0}^{X_d}$ and $\{x_i'\}_{i=0}^{X_d}$, of the same size $X_d$. Then **DIV** is formalized as:

$$\text{DIV} = \frac{1}{X_d} \Sigma_{i=1}^{X_d} \|x_i - x_i'\| \tag{9}$$

Randomly sample a set of text descriptions with size $J_m$ and sample two subsets of $X_m$ motions generated by $j$-th for each text description, denote as $\{x_{j,i}\}_{i=0}^{X_m}$ and $\{x_{j,i}'\}_{i=0}^{X_m}$. **MM** is calculated as:

$$\text{MM} = \frac{1}{J_m \times X_m} \Sigma_{i=1}^{J_m} \Sigma_{i=1}^{X_m} \|x_{j,i} - x_{j,i}'\| \tag{10}$$

**Motion captioning**   We follow prior work chuan2022tm2t and adopt standard NLP metrics including BLEU (Papineni et al., 2002), ROUGE-L (Lin, 2004), CIDEr (Vedantam et al., 2015), and BERTScore (Zhang et al., 2019) to evaluate the fluency, relevance, and diversity of generated captions.

**Reconstruction**   We evaluate reconstruction fidelity of motion autoencoders with: **MPJPE** and **PAMPJPE** for global and local errors in millimeter, **ACCL** (Acceleration Error) computed from second-order finite differences, and **APE/AVE** (Absolute Position/Velocity Error) reported over root, trajectory, pose, and joints components.

## E   THE USE OF LARGE LANGUAGE MODELS (LLMS)

We used ChatGPT only for grammar/typo checks; all technical content, experiments, and analyses were authored and verified by the authors without substantive LLM contribution.

