# OpenReview forum: "MotionGPT3: Human Motion as a Second Modality"
_ICLR.cc/2026/Conference — ICLR 2026 Poster_

### Official Review · Reviewer_9qBu · 2025-10-30

**Soundness:** 3
**Presentation:** 3
**Contribution:** 3
**Rating:** 6
**Confidence:** 3

**Summary:**

This paper presents MotionGPT3, a motion–language model designed to jointly handle motion understanding and generation while addressing key limitations of existing multimodal frameworks. To avoid the motion quantization issue, it encodes raw motion into a continuous latent space with a lightweight diffusion head, eliminating quantization-induced artifacts and enabling higher-fidelity synthesis. The architecture employs a dual-stream Transformer with shared attention, which maintains modality-specific information while allowing cross-modal exchange. Besides, a three-stage generate-then-align training schedule is proposed to further enhance the  convergence  efficiency.

**Strengths:**

* The paper is well written, easy to follow, and clearly organized. The figures are self-explanatory.
* The motivation is concrete and reasonable, and the method design is aligned well with the corresponding motivations, presenting sound performance improvements.
* This paper offers a well-reasoned perspective on the gap between discrete language token sequences and continuous motion latent representations, which can provide valuable insights to the community.

**Weaknesses:**

* Lack of detailed comparisons of inference latency or FLOPs against discrete-token baselines. Authors should provide the computational overheads induced by each proposed component to demonstrate the method's efficiency.
* The evaluation of the method is constrained to a single dataset (HumanML3D), which can not sufficiently demonstrate the generalization of the proposed framework to other motion domains. Providing more results on other benchmarks with diverse features would better illustrate the method's versatility. Besides, it is expected to offer more results with more recent language models other than GPT-2.
* As shown in Table 1, why does the proposed method significantly lower on the MModality metric compared to other models? Does this indicate that the model can only fits a specific distribution, thereby sacrificing the diversity of generated outputs.

**Questions:**

* typos: line 32, citepacross

---

> ### Author Response · Authors · 2025-11-24
> **Reply to Reviewer 9qBu (1/2)**
>
> We thank the reviewer for the constructive feedback. We have addressed each weakness and question in the corresponding point-wise responses.
>
> We provide (i) additional experiments on more datasets and language backbones, (ii) detailed inference-latency comparisons and component-wise computational analysis, and (iii) clarification of the MModality behavior and its relation to overall diversity and alignment. We also corrected the noted typo.
>
> We hope these additions effectively address the reviewer’s concerns and further strengthen the presentation of our method.
>
> ---
> ### W2. More results on other benchmarks with diverse features. More results with more recent language models.
>
> > W2. The evaluation of the method is constrained to a single dataset (HumanML3D), which can not sufficiently demonstrate the generalization of the proposed framework to other motion domains. Providing more results on other benchmarks with diverse features would better illustrate the method's versatility. Besides, it is expected to offer more results with more recent language models other than GPT-2.
>
>
> - **On more recent language models.**
>
>   To address the concern about using only GPT-2, we additionally instantiate our framework with two more recent language backbones, Qwen2.5-0.5B-Instruct and Flan-T5-base, under both single- and dual-stream architectures.
>
>   All single-stream models are trained for 400 epochs and dual-streamm models for 200 epochs. For each setting, we report the checkpoint with the lowest validation FID on HumanML3D. The results are summarized below.
>
>   **Table R7:** T2M generation on backbones of Qwen2.5, T5, and GPT2, experiments are conducted on both single- and dual-stream architectures. The results are sorted by model size (Parameters).
>
>   | Backbone | Architecture | Parameters | R@1       | R@2       | R@3       | FID↓      | MMDist↓   | Diversity↔ |
>   | -------- | --------- | -----|---- | ------ | --------- | --------- | --------- | ---------- |
>   | Real     | -          | - | 0.511     | 0.703     | 0.797     | 0.002     | 2.974     | 9.503      |
>   | GPT2     | single |   152M  | 0.513     | 0.708  | 0.808 | 0.485  | 2.962 | **9.633** |
>   | GPT2     | dual | 238M| *0.533*   | *0.731*   | *0.826*   | **0.239** | *2.797*   | 9.688      |
>   | T5       | single  | 275M  | 0.491     | 0.681     | 0.777     | *0.243*   | 3.102     | *9.365*  |
>   | T5       | dual    | 369M | 0.509     | 0.702     | 0.799     | 0.285     | 2.978     | 9.772      |
>   | Qwen2.5  | single |  524M  | 0.483     | 0.669     | 0.768     | 0.491     | 3.143     | 9.653    |
>   | Qwen2.5  | dual    |  617M | **0.542** | **0.739** | **0.832** | 0.309     | **2.782** | 9.878      |
>
> Across all three backbones (GPT-2, Qwen, and T5), introducing our dual-stream architecture consistently improves retrieval metrics (R@1/2/3) and MMDist over the corresponding single-stream variants, demonstrating that our framework is not tied to a particular language model and scales well to more recent LMs.
>
>
> - **On additional benchmarks.**
>
>   To further evaluate generalization beyond HumanML3D, we also conduct experiments on the KIT-ML dataset. We train a VAE-based model and the corresponding T2M model (MotionGPT3$\dagger$) and report:
>
>   **Table R8:** Motion generation on KIT-ML dataset. We also report the reconstructioin metrics for VAE used in MotionGPT3$\dagger$.
>
>   | Methods   | R@1↑    | R@2↑    | R@3↑   | FID↓  | MMDist↓   | Diversity↔ | MModality↑ |
>   | ---- | --- | ---- | --- | ---- | ---- | ----- | ----- |
>   | GT      | 0.424     | 0.649     | 0.779     | 0.031     | 2.788     | 11.08      | -      |
>   | VAE         | 0.4085 | 0.6273 | 0.7626 | 0.1548 | 2.855 | 10.9193 | -      |
>   | MotionGPT3$\dagger$ | **0.4549**| **0.6795**| **0.8027**| 0.2271| **2.7043** | *11.0255* | 0.9036   |
>   | MLD           | 0.390     | 0.609     | 0.734     | 0.404     | 3.204   | 1.080    | 2.192    |
>   | MotionGPT     | 0.366     | 0.558     | 0.680     | 0.510     | 3.527     | 10.350     | *2.328*    |
>   | MotionDiffuse    | 0.417   | 0.621   | 0.739   | 1.954   | 2.958   | **11.100** | 0.730    |
>   | ReMoDiffuse       | 0.427   | 0.641   | 0.765   | *0.155*   | 2.814   | 10.800     | 1.239    |
>   | MoMask        | 0.433   | 0.656   | 0.781   | 0.204     | 2.779     | -      | 1.131      |
>   | MoTe         | 0.419   | 0.627   | 0.741   | 0.256     | 3.216     | -      | **2.615**  |
>   | MoGenTS      | *0.445* | *0.671* | *0.797* | **0.143** | *2.711* | 10.918     | -     |

---

> ### Author Response · Authors · 2025-11-24
> **Reply to Reviewer 9qBu (2/2)**
>
> ### W1. inference latency
>
> > W1. Lack of detailed comparisons of inference latency or FLOPs against discrete-token baselines. Authors should provide the computational overheads induced by each proposed component to demonstrate the method's efficiency.
>
> To address the concern on inference latency, we compared the inference times for different architectures: VQ + single-stream, VAE + single-stream, and VAE + dual-stream, across both T2M and M2T tasks. We measured the time for key sub-processes such as encoding, decoding, and backbone generation.
>
> For evaluation, we test on 100 data with a batch size of 1 and reported the average time per task across all models. Our results show that VAE and diffusion head significantly improve *efficiency for T2M task* , while maintaining competitive M2T performance.
>
>
> **Table R6:** Detailed inference time on different architectures across both T2M and M2T tasks.
>
> | Model      | T2M Total | M2T Total | T2M Generation | T2M Diffusion | T2M Decode | M2T Encode | M2T Generation |
> | ------- | ------ | ----- | ---- | ----- | ---- | ----- | ---------- |
> | VQ                     | 384.50    | 166.19    | 389.23         | -             | 14.01      | 18.08      | 146.58         |
> | VAE                    | 172.53    | 393.39    | 12.72          | 141.86        | 8.62       | 10.45      | 413.38         |
> | MoT + VAE (Ours) | 202.80    | 226.95    | 42.00          | 158.81        | 9.76       | 11.35      | 224.86         |
> | MotionGPT              | 489.18    | 231.16    | 476.61         | -             | 13.01      | 5.45       | 228.01         |
>
>
> ---
>
> ### W3. lower on the MModality metric compared to other models
>
> > W3. As shown in Table 1, why does the proposed method significantly lower on the MModality metric compared to other models? Does this indicate that the model can only fits a specific distribution, thereby sacrificing the diversity of generated outputs.
>
> We thank the reviewer for highlighting this point. We clarify that our model does **not** sacrifice motion diversity.
>
> **First**, in terms of *overall diversity*, our method achieves Diversity scores on par with or higher than strong baselines (e.g., 9.700 vs. 9.620 for MoMask). This shows that our model covers a broad range of motions and does not collapse to a narrow distribution.
>
> **Second**, the MModality metric measures the variation *among multiple samples generated from the same text input*, computed inside the embedding space of the T2M evaluator. A lower MModality in our model does not imply reduced generative diversity, but rather that for a given text prompt, the samples are more semantically consistent. This behavior aligns with our stronger text–motion alignment (higher R-precision, lower MMDist).
>
> **Third**, we observe a similar trend in strong recent models (e.g., MoMask, ReMoDiffuse): as conditional quality improves (higher RPrecision/ lower MMDist), MModality often decreases. This suggests that high-fidelity, text-consistent motion generation tends to produce tighter conditional clusters in the evaluator’s embedding space.
>
> **Finally**, we are aware that current metrics do not fully disentangle conditional diversity from alignment quality. Developing more reliable measures for multi-modal generation is an important direction for future work.
>
> ---
> ### **Q1. typos**
>
> > Q1. typos: line 32, citepacross
>
> We thank the reviewer for pointing this out and have corrected the typo in the revised manuscript.

---

> > ### Comment · Reviewer_9qBu · 2025-11-26
> >
> > Thanks to the authors for providing more comprehensive results regarding the LMs and additional benchmarks. Most of my concerns have been well addressed. However, the proposed MotionGPT3 does not achieve the same level of superior performance on the additional T2M benchmark. Thus, I still have concerns about the generalizability of the framework. Overall, this paper presents a sound exploration of unifying motion understanding and generation. I am inclined to maintain my original positive score.

---

> ### Author Response · Authors · 2025-11-26
> **Response to reviewer’s concern on generalizability across KIT benchmark**
>
> Thank you for your timely feedback. In our previous response, due to time constraints, we only shared our *initial* results on the KIT-ML dataset, which showed moderate performance due to an under-optimized VAE.
> So far, we have fine-tuned the VAE on KIT-ML, which has led to significant improvements in subsequent generation tasks, surpassing state-of-the-art methods. For example, compared to MoGenTS, our model achieved **R@1 of 0.4549** (vs. 0.445) and **MMDist of 2.7043** (vs. 2.711).
> The updated results, now shown in Table R3, demonstrate the effectiveness of these optimizations and further highlight the generalizability of our framework.

---

### Official Review · Reviewer_cYVn · 2025-10-31

**Soundness:** 3
**Presentation:** 3
**Contribution:** 2
**Rating:** 4
**Confidence:** 4

**Summary:**

This paper presents MotionGPT3, a unified model that addresses motion quantization errors and cross-modal interference in motion-language tasks. Its key innovations include: 1) a VAE-based continuous motion latent space, 2) a dual-stream Transformer for controlled cross-modal interaction, and 3) a diffusion head to connect discrete text and continuous motion. Trained in three stages, MotionGPT3 sets new state-of-the-art performance on HumanML3D for both text-to-motion and motion-to-text generation, while converging 2-4 times faster than baselines.

**Strengths:**

1. The technical design in this paper is highly targeted, with each core component—a continuous VAE, a dual-stream architecture, and three-stage training—precisely solving a specific problem, and its necessity is rigorously validated through ablation studies, resulting in a lean and non-redundant overall architecture.

2. The experimental validation is comprehensive, encompassing quantitative comparisons, qualitative examples, and thorough ablation studies.

3. The study ensures high reproducibility by detailing implementation specifics—including data preprocessing, training protocols, and evaluation tools—and adheres to open-source standards with released code and materials.

**Weaknesses:**

1. The bimodal branch architecture proposed in this paper to address cross-modal interference in motion-language modeling is not a particularly novel approach, as similar frameworks have been proposed in existing unified text-image understanding and generation work, such as BAGEL [1]. However, the paper lacks discussion on how the proposed method differs from these existing approaches.

2. Baseline comparisons are outdated, lacking recent models (e.g., MotionGPT-2 2024 [2]，MG-MotionLLM 2025 [3]), which may conceal performance gaps in key metrics like M2T BERTScore.

3. The paper provides analysis on training convergence speed but lacks evaluation of inference latency, which remains a critical metric for motion generation applications.

[1] Emerging Properties in Unified Multimodal Pretraining, 2025, arxiv, 27 July
[2] MotionGPT-2: A General-Purpose Motion-Language Model for Motion Generation and Understanding, 2024 arxiv, 29 Oct
[3] MG-MotionLLM: A Unified Framework for Motion Comprehension and Generation across Multiple Granularities, 2025 arxiv, 3 April

**Questions:**

Please see the weakness part.

---

> ### Author Response · Authors · 2025-11-24
> **Reply to reviewer cYVn (1/2)**
>
> We thank the reviewer for the positive assessment of our writing, motivation, and perspective on continuous motion representations.
>
> Regarding the concern you raised, we (i) added a clearer discussion on how our bimodal architecture differs from prior unified text–image frameworks such as BAGEL, with an emphasis on motion-specific design and objectives, (ii) extended our experimental comparison to include recent baselines such as MotionGPT-2 on both T2M and M2T (the  comparison to MG-MotionLLM on T2M was presented in Fig. 8 of the submitted paper), and (iii) reported inference latency for different architectures and motion representations, breaking down the main components of the decoding pipeline.
>
> We hope these clarifications and additional results help convey the distinct contribution and practical relevance of our method.
>
> ---
> ### W1. Novelty & Motivation.
>
> > W1. The bimodal branch architecture proposed in this paper to address cross-modal interference in motion-language modeling is not a particularly novel approach, as similar frameworks have been proposed in existing unified text-image understanding and generation work, such as BAGEL [1]. However, the paper lacks discussion on how the proposed method differs from these existing approaches.
>
> While our bimodal architecture shares similarities with text-image models like BAGEL, our work is *focused exclusively on motion-related tasks*.
>
> The key difference between our bimodal architecture and the dual-branch framework of BAGEL is that our dual-stream are *modality-wise* while the branches in BAGEL are *task-specific*, i.e., one branch for understanding and one branch for generation, with seperate experts and adaptors. Other detailed differences include:
>
> - **Focused on Motion.**
>   Unlike text-image models, our approach is optimized specifically for motion generation and understanding, incorporating motion-specific innovations like *VAE representation* and *feedforward motion generation*.
>
> - **Modality Separation and Alignment.**
>   We introduce a *modality-based weight separation* and focus on representation alignment to ensure coherent and accurate cross-modal learning between text and motion.
>
> - **Motion-Specific Optimizations.**
>   We also use *special tokens* for interleaved text-motion sequences and a *lightweight diffusion module* to balance performance and efficiency.
>
> These design choices are tailored to motion generation, making our approach distinct from text-image models.
>
> ---
> ### W3. evaluation of inference latency
>
> > W3. The paper provides analysis on training convergence speed but lacks evaluation of inference latency, which remains a critical metric for motion generation applications.
>
> To address the concern on inference latency, we compared the inference times for different architectures: VQ + single-stream, VAE + single-stream, and VAE + dual-stream, across both T2M and M2T tasks. We measured the time for key sub-processes such as encoding, decoding, and backbone generation.
>
> For evaluation, we test on 100 data with a batch size of 1 and reported the average time per task across all models. Our results show that **VAE and diffusion head** significantly improve **efficiency for** **T2M task** , while maintaining competitive M2T performance.
>
>
>
> **Table R12:** Detailed inference time on different architectures across both T2M and M2T tasks.
>
> | Model     | T2M Total | M2T Total | T2M Generation | T2M Diffusion | T2M Decode | M2T Encode | M2T Generation |
> | --------- | --------- | --------- | -------------- | ------------- | ---------- | ---------- | -------------- |
> | VQ        | 384.50    | 166.19    | 389.23         | -             | 14.01      | 18.08      | 146.58         |
> | VAE       | 172.53    | 393.39    | 12.72          | 141.86        | 8.62       | 10.45      | 413.38         |
> | MoT + VAE | 202.80    | 226.95    | 42.00          | 158.81        | 9.76       | 11.35      | 224.86         |
> | MotionGPT | 489.18    | 231.16    | 476.61         | -             | 13.01      | 5.45       | 228.01         |

---

> ### Author Response · Authors · 2025-11-24
> **Reply to reviewer cYVn (2/2)**
>
> ### W2. Recent baselines.
>
> > W2. Baseline comparisons are outdated, lacking recent models (e.g., MotionGPT-2 2024 [2]，MG-MotionLLM 2025 [3]), which may conceal performance gaps in key metrics like M2T BERTScore.
>
> In our paper, we compare our method with several recent, high-performing works such as MoTe, MG-MotionLLM, and LaMP, focusing on both T2M and M2T metrics. The relevant results are shown in Table 1 and Table 2 in the main text, and additional T2M comparisons are provided in Appendix Sec. C.1, with Table 6 for single-task T2M methods, and Figure 8 for unified tasks.
>
> For convenience, we present a comparison of the methods mentioned by the reviewer, along with other recent models from 2024 onwards, in the tables below.
>
>
>
> **Table R9:** T2M Comparison of recent approaches on HumanML3D.
>
> | Methods             | Top1      | Top2      | Top3      | FID↓      | MMDist↓   | Diversity↔ | MModality↑ |
> | :------------- | ------ | ------ | ------- | ------- | ------- | -------- | -------- |
> | Real                | 0.511     | 0.703     | 0.797     | 0.002     | 2.974     | 9.503      | -          |
> | MotionDiffuse       | 0.491     | 0.681     | 0.782     | 0.630     | 3.113     | 9.410      | 1.553      |
> | MotionGPT-2         | 0.496     | 0.691     | 0.782     | 0.191     | 3.080     | 9.860      | 2.137      |
> | MoMask              | 0.521     | 0.713     | 0.807     | 0.045     | 2.958     | 9.620      | 1.241      |
> | MoTe                | 0.548     | 0.737     | 0.825     | 0.075     | 2.867     | -          | *2.399*    |
> | MotionAnything      | 0.546     | 0.735     | 0.829     | **0.028** | 2.859     | **9.521**  | **2.705**  |
> | LaMP                | **0.557** | **0.751** | **0.843** | *0.032*   | *2.759*   | *9.571*    | -          |
> | MG-MotionLLM        | 0.516     | 0.706     | 0.802     | 0.303     | 2.952     | 9.960      | 2.125      |
> | MotoinGPT3$\dagger$ | 0.533     | 0.731     | 0.826     | 0.239     | 2.797     | 9.688      | 1.560      |
> | MotoinGPT3          | *0.553*   | *0.747*   | *0.837*   | 0.208     | **2.725** | 9.700      | 1.018      |
>
>
> **Table R10:** M2T Comparison of recent approaches on HumanML3D.
>
> | Methods    | R@1    | R@2   | R@3   | MMDist↓   | Bleu@1↑    | Bleu@4↑    | Rouge↑     | Cider↑   | BertScore↑ |
> | ---------- | ------ | ----- | ---- | ----- | ---- | ----- | ------ | ----- | ------ |
> | MotioGPT-2          | 0.558     | 0.738 | 0.838     | 2.767     | 48.7       | 13.8       | 37.6       | 29.8     | 32.6       |
> | MoTe                | *0.577*   | -     | **0.871** | 2.649     | 46.7       | 11.15      | 37.4       | **31.5** | 30.3       |
> | LaMPM2T             | 0.547     | -     | 0.831     | 2.808     | 47.8       | 13.04      | 37.1       | 28.9     | 32.7       |
> | MG-MotionLLM        | **0.592** | -     | *0.866*   | 2.581     | -          | 8.06       | -          | -        | **36.7**   |
> | MotoinGPT3$\dagger$ | 0.553     | 0.756 | 0.853     | *2.524*   | *56.363*   | *17.661*   | *44.997*   | *30.980* | *35.850*   |
> | MotoinGPT3          | 0.573     | 0.773 | 0.864     | **2.426** | **59.083** | **19.412** | **46.173** | 28.721   | 35.231     |
>
> Our method shows strong performance in both T2M and M2T tasks, outperforming previous methods in many metrics.
>
> ---
> Regarding the observation that some SOTA methods *outperform GT* in both motion generation and motion captioning tasks, we believe this may suggest that the current evaluation methods (such as the T2M evaluator) may not fully capture the high-quality generated motions.
>
> We discuss this in Appendix C.2 and provide additional TMR metrics there. While our method slightly underperforms LaMP in T2M metrics, we achieve superior results in TMR evaluation.
>
>
> **Table R11:** Text-to-motion comparison on “small batch” protocol of TMR evaluator.
>
> **(a) Text-to-Motion Retrieval**
>
> | Methods   | R@1↑    | R@2↑    | R@3↑    | R@5↑    | R@10↑   | MedR↓  |
> | ---- | ----- | ---- | --- | --- | ---- | --- |
> | T2M       | 52.48     | 71.05     | 80.65     | 89.66     | 96.58     | 1.39     |
> | TMR       | 67.16     | 81.32     | 86.81     | 91.43     | 95.36     | 1.04     |
> | MotionGPT | 58.07     | 69.91     | 74.34     | 79.17     | 86.36     | 1.18     |
> | LaMP      | 67.18     | 81.90     | 87.04     | 92.00     | 95.73     | -        |
> | MotionGPT3      | **74.25** | **86.70** | **91.29** | **94.82** | **97.35** | **1.00** |
>
> **(b) Motion-to-Text Retrieval**
>
> | Methods  | R@1↑   | R@2↑  | R@3↑  | R@5↑  | R@10↑     | MedR↓    |
> | --- | --- | ---- | --- | --- | -- | --- |
> | T2M       | 52.00     | 71.21 | 81.11     | 89.87 | 96.78     | 1.38     |
> | TMR       | 67.97     | 81.20 | 86.35     | 91.70 | 95.27     | 1.03     |
> | MotionGPT | 58.71     | 69.64 | 74.36     | 79.45 | 86.02     | 1.16     |
> | LaMP      | 68.02     | 82.10 | 87.50     | 92.20 | 96.90     | -        |
> | MotionGPT3      | **74.00** | **86.86** | **91.04** | **94.62** | **97.35** | **1.00** |

---

### Official Review · Reviewer_VYUy · 2025-11-01

**Soundness:** 2
**Presentation:** 3
**Contribution:** 2
**Rating:** 4
**Confidence:** 5

**Summary:**

This paper propose MotionGPT3, a bimodal motion-language framework to unify the motion understanding and generation. It introduces two different modality branches to endow the model with motion-to-text and text-to-motion abilities.

**Strengths:**

1. The paper contains numerous figures and tables, as well as abundant visualization results, with a relatively clear overall structure.

2. Video demos are provided, demonstrating excellent performance.

3. Motion generation and motion understanding tasks are realized through two different branches and fine-tuning of the LLM.

4. The experimental results have achieved significant improvements.

**Weaknesses:**

1. The description of Figure 2 and the method section is not clear enough, making it difficult to intuitively grasp the authors' entire training process design and the detailed reasoning procedure.

2. Although the paper achieves good results, the method feels relatively incremental and highly hierarchical, lacking overall simplicity. Compared with MotionGPT and MotionGPT2, it does not bring a strong sense of novelty.

3. The autoregressive continuous token proposed by the authors has been used in many motion generation papers, which makes the contribution seem insufficient.

4. There are some typos. For example, is "L40" supposed to be "LLM"? The content from Line 82 to Line 89 is better presented in bullet points. Rarely seen in introductions is the writing style from Line 57 to Line 80, which fails to convey the necessity and practicality of the method proposed by the authors.

**Questions:**

1. It is hoped that the authors can elaborate on the training details of the three stages, including the inputs and outputs. In particular, they should clarify why such inputs are used in the second stage and what advantages they bring. Additionally, what is the main purpose of keeping the text branch frozen in the second stage? Could the authors provide an input example for Stage 2?

2. What exactly are the advantages of MotionGPT3 compared to MotionGPT and MotionGPT2, and which problems that the latter two failed to solve have been addressed? Relevant experiments would be highly appreciated.

3. The authors are expected to clarify the points mentioned in the "Weakness" section.

If the authors can address the questions I raised, I may consider increasing the score.

---

> ### Author Response · Authors · 2025-11-24
> **Reply to Reviewer VYUy (1/2)**
>
> We thank the reviewer for the positive assessment of our results, visualizations, and overall structure. We appreciate the constructive feedback as well and have carefully addressed each concern in the point-wise responses. We provide clearer explanations of the training pipeline and Figure 2, clarify how our approach differs from prior approaches leveraging autoregressive continuous token, and also address the questions regarding MotionGPT3’s advantages. We hope these clarifications help convey the method’s rationale, novelty, and technical contributions more clearly.
>
> **First of all,** Regarding novelty concerns, while continuous latent modeling with autoregressive backbones has appeared in some concurrent motion-generation works, our contribution lies in building a unified motion–language model with modality-separated branches, together with targeted design choices (continuous VAE latent space, lightweight diffusion head, interleaved special tokens), and a three-stage training strategy that significantly improves cross-modal alignment and joint T2M/M2T performance.
>
> We kindly suggest the reviewer refer to the detailed explanation of our motivation and novelty provided in the *Overall Clarification* section.
>
> ---
> ### W1&Q1. training details of the three training stages and inference settings.
>
> > Q1. It is hoped that the authors can elaborate on the training details of the three stages, including the inputs and outputs. In particular, they should clarify why such inputs are used in the second stage and what advantages they bring. Additionally, what is the main purpose of keeping the text branch frozen in the second stage? Could the authors provide an input example for Stage 2?
>
> **1. Special Tokens and Supervision**
>
> We detail the supervision mechanism and special tokens in Sec. D.1. For all tasks, we use `<motion_in>` / `<motion_out>`to mark motion positions in the input/output sequences.
>
> - **Motion as input**: we encode the motion latent and pass it through an match the model’s input dimension, and replace the embeddings at `<motion_in>` with this adapted latent.
> - **Motion as output**: we use the hidden states at `<motion_out>` positions as the conditioning for the diffusion head, which denoises to recover the motion latent.
>
> Text tokens are supervised as in standard LLM training, while motion latents follow a DDIM-style diffusion objective.
>
> **2. Training Stages**
>
> - **Stage1:** We freeze the text branch and train only on the T2M task. The input is a text sequence followed by K `<motion_out>`tokens. The model produces K hidden states at these positions, which are used to condition the diffusion head to denoise towards the ground-truth motion latents.
>
> - **Stage2:** The text branch remains frozen, and we train on both T2M and M2T. For M2T, the input contains K `<motion_in>` tokens. the VAE-encoded motion latent, is passed through an adaptor and used to replace the embeddings at these positions. In this stage, the motion branch learns motion understanding and is encouraged to align with the pretrained text branch, which is why we keep the text branch frozen.
>
> - **Stage 3:** We unfreeze both branches and jointly fine-tune on T2M and M2T. This stage refines both generation and understanding and further improves motion–language alignment.
>
> **3. Inference process**
>
> - **M2T**: we encode the motion to a latent, insert it at the `<motion_in>` position via the adaptor, and then generate text as in a standard LLM.
>
> - **T2M**: After the text branch predicts the `<som>` token, we insert K `<motion_out>` tokens, obtain the corresponding hidden states in one forward pass, and feed them into the diffusion head to generate the motion latent.
>
> ---
> ### W4. Typos
>
> > There are some typos. For example, is "L40" supposed to be "LLM"? The content from Line 82 to Line 89 is better presented in bullet points. Rarely seen in introductions is the writing style from Line 57 to Line 80, which fails to convey the necessity and practicality of the method proposed by the authors.
>
> - **"LM" in L40**:
>
>   We thank the reviewer for pointing this out and have corrected the typo in the revised manuscript.
>
> - **Bullet points for contributions**:
>
>   We agree that presenting the contributions in bullet points will improve clarity. We will update the final version accordingly.
>
> - **Writing style (Line 57 to Line 80)**:
>
>   The writing style in this section was inspired by the motivational approach used in works like Stable Diffusion, aiming to explain the theoretical and practical motivations behind the method.
>
>   We will revise this section to make the necessity and practicality of our approach more explicit for the reader.

---

> ### Author Response · Authors · 2025-11-24
> **Reply to Reviewer VYUy (2/2)**
>
> ### W2. & Q2. advantages compared to MotionGPT and MotionGPT2
>
> > W2. Although the paper achieves good results, the method feels relatively incremental and highly hierarchical, lacking overall simplicity. Compared with MotionGPT and MotionGPT2, it does not bring a strong sense of novelty.
> >
> > Q2. What exactly are the advantages of MotionGPT3 compared to MotionGPT and MotionGPT2, and which problems that the latter two failed to solve have been addressed? Relevant experiments would be highly appreciated.
>
> - **Design differences.**
>
>   MotionGPT and MotionGPT2 both follow the approach of tokenizing motion into discrete tokens and leveraging a single-stream backbone for learning both modalities, enabling motion understanding and generation.
>
> - **Key improvement in MotionGPT3.**
>
>   In contrast, MotionGPT3 introduces continuous VAE latent as motion representation, and a **dual-stream framework**which effectively mitigates interference between modalities, stabilizing the training process. These design results in better performance, with improved generation and understanding metrics.
>
> - **Empirical evidence.**
>
>   Our experiments demonstrate consistent improvements over MotionGPT and MotionGPT-2, including better R-precision, MMDist, Bleu, BertScore, and faster convergence.
>
>   **Table R4:** T2M comparison on HumanML3D dataset.
>
>   | Methods             | Top1      | Top2      | Top3      | FID↓      | MMDist↓   | Diversity↔ | MModality↑ |
>   | ------------------- | --------- | --------- | --------- | --------- | --------- | ---------- | ---------- |
>   | Real                | 0.511     | 0.703     | 0.797     | 0.002     | 2.974     | 9.503      | -          |
>   | MotionGPT           | 0.492     | 0.681     | 0.733     | 0.232     | 3.096     | **9.528**  | 2.000      |
>   | MotionGPT-2         | 0.496     | 0.691     | 0.782     | **0.191** | 3.080     | 9.860      | **2.137**  |
>   | MotionGPT3$\dagger$ | 0.533     | 0.731     | 0.826     | 0.239     | 2.797     | 9.688      | 1.560      |
>   | MotionGPT3          | **0.553** | **0.747** | **0.837** | 0.208     | **2.725** | 9.700      | 1.018      |
>
>
>
>   **Table R5:** M2T comparison on HumanML3D dataset.
>
>   | Methods             | R@1       | R@2       | R@3       | MMDist↓   | Bleu@1↑    | Bleu@4↑    | Rouge↑     | Cider↑     | BertScore↑ |
>   | ------------------- | --------- | --------- | --------- | --------- | ---------- | ---------- | ---------- | ---------- | ---------- |
>   | Real                | 0.523     | 0.725     | 0.828     | 2.901     | -          | -          | -          | -          | -          |
>   | MotioGPT            | 0.543     | -         | 0.827     | 2.821     | 48.2       | 12.5       | 37.4       | 29.2       | 32.4       |
>   | MotioGPT-2          | 0.558     | 0.738     | 0.838     | 2.767     | 48.7       | 13.8       | 37.6       | 29.8       | 32.6       |
>   | MotionGPT3$\dagger$ | 0.553     | 0.756     | 0.853     | 2.524     | 56.363     | 17.661     | 44.997     | **30.980** | **35.850** |
>   | MotionGPT3          | **0.573** | **0.773** | **0.864** | **2.426** | **59.083** | **19.412** | **46.173** | 28.721     | 35.231     |
>
>
> ---
> ### W3. Discussion on recent works with continuous token autoregressive.
>
> > W3. The autoregressive continuous token proposed by the authors has been used in many motion generation papers, which makes the contribution seem insufficient.
>
> We agree that continuous latent/token autoregression is an emerging direction, with only a **few very recent concurrent works** exploring related ideas. These methods, however, mostly target specific scenarios such as streaming generation [1] or keyframe-based control[2], rather than unified motion–language modeling.
>
> Our contribution goes beyond simply combining continuous tokens with an autoregressive backbone, but designing a **unified dual-stream motion–language model** that (i) supports interleaved text–motion inputs, (ii) enables bidirectional generation within a single architecture, and (iii) explicitly improves cross-modal alignment. To our knowledge, such a unified LLM-style framework operating on continuous motion tokens has not been explored in prior work.
>
> [1] Xiao, Lixing, et al. MotionStreamer: Streaming Motion Generation via Diffusion-based Autoregressive Model in Causal Latent Space. *arXiv preprint arXiv:2503.15451* (2025).
>
> [2] Zhang, Zongye, et al. Towards robust and controllable text-to-motion via masked autoregressive diffusion. *Proceedings of the 33rd ACM International Conference on Multimedia*. 2025.

---

### Official Review · Reviewer_Hcm9 · 2025-11-05

**Soundness:** 3
**Presentation:** 3
**Contribution:** 3
**Rating:** 6
**Confidence:** 4

**Summary:**

This paper propose a bimodal motion–language model for text-to-motion and motion-to-text generation. The key designs include: a continuos latent modtion space with a latent diffusion header to bridge its gap between the next-token predition framework, a dual branch framework with shared attention to bridge the gap of the two modalities, and a three stage training strategy for more stable optimization of the proposed framework. Experimental results show the effectiveness of the proposed framework on both the text-to-motion and motion-to-text tasks.

**Strengths:**

- The motivation of utilizing the continuous motion latent space for lossless motion encoding and the diffusion header to bridge the gap between the next-token generation framework is reasonable.
- The dual-branch framework to preserve modality-specific information and the shared attention for cross-modal communication is well motivated, and the three-stage training schemes stabilize the optimization of the proposed framework.
- Experimental results on benchmarks of the two tasks are strong, and the effect of different design choices is validated with ablation studies.
- The writing is good and the paper is easy to understand.

**Weaknesses:**

- The paper claims continuous VAE for motion encoding is better, but lacks an experimental comparison on motion encoding and decoding quality with previous schemes. Specifically, how is the improvement of the continuous VAE compared to the recently stronger motion quantization methods, e.g., the residual VQ proposed by MoMask (CVPR 2024) and the 2D motion quantization in MoGenTS (NeurIPS 2024)?
- Experiments are only conducted on the HumanML3D datasets. Adding more diverse datasets, e.g., Motion-X and  KIT-ML, will better illustrate the generalizability of the proposed framework.

**Questions:**

None.

---

> ### Author Response · Authors · 2025-11-24
> **Reply to Reviewer Hcm9 (1/2)**
>
> We thank the reviewer for the positive remarks on the motivation, model design, and empirical validation.
>
> The concerns raised are also well taken. In our point-wise responses, we provide (i) comparisons between our continuous VAE representation and recent quantization-based motion encoders, and (ii) additional experiments on broader benchmarks to assess generalization beyond HumanML3D. We hope these clarifications and extended analyses further substantiate the effectiveness and generability of our framework.
>
> ---
> ### W1. Comparison on motion encoding and decoding quality with previous schemes. continuous VAE v.s. recently stronger motion quantization methods
>
> > W1. The paper claims continuous VAE for motion encoding is better, but lacks an experimental comparison on motion encoding and decoding quality with previous schemes. Specifically, how is the improvement of the continuous VAE compared to the recently stronger motion quantization methods, e.g., the residual VQ proposed by MoMask (CVPR 2024) and the 2D motion quantization in MoGenTS (NeurIPS 2024)?
>
> We thank the reviewer for the insightful question. We provide reconstruction comparison between our continuous VAE (MLD), and recent quantization-based schemes, including VQVAE (MotionGPT), RVQVAE (MoMask), and 2D motion quantization (MoGenTS):
>
> **Table R1:** Comparison on reconstruction quality for VQVAE, MLDVAE, RVQVAE, and 2D VQ. Evaluated with protocol from MoMask.
>
> | Method            | FID↓        | MPJPE↓ | RTOP3↑      | MMDist↓     |
> | ----------------- | ----------- | ------ | ----------- | ----------- |
> | VQVAE (MotionGPT) | 0.061 | 58.0   | 0.780 | 3.094 |
> | MoMask (RVQVAE)   | 0.019 | 29.5   | 0.793 | 3.008 |
> | MoGenTS (2D VQ)   | 0.005 | 15.8   | 0.799 | 2.973 |
> | Ours (MLDVAE)     | 0.171  | 43.9   | 0.784  | 3.070  |
>
> As shown, recent quantization-based models such as MoMask and MoGenTS achieve very strong reconstruction performance.
> We anticipate that the weaker reconstruction results compared to RVQVAE and 2D VQ may due to the *unoptimized MLD-VAE*, since we adopt a *vanilla* MLD-VAE with a latent size of only 1×256 in all experiments.
> We acknowledge that stronger reconstruction models may further benefit generation quality. Therefore, we are currently working on training an enhanced MLD-VAE with stronger reconstruction capability and conducting subsequent experiments.
>
>
> **Nevertheless**, our work targets generation quality and multimodal alignment, rather than reconstruction alone. On the standard T2M generation benchmark, our method achieves stronger metrics on text–motion alignment (R-Precision, MMDist), indicating that the proposed framework provides clear advantages for both generation and cross-modal alignment.
>
>
> **Table R2:** Comparison on T2M for MoMask, MoGenTS, MotionGPT, and MotionGPT3.
>
> | Methods   | R@1↑      | R@2↑      | R@3↑      | FID↓      | MMDist↓   | Diversity↔ |
> | --------- | --------- | --------- | --------- | --------- | --------- | ---------- |
> | MoMask    | 0.521     | 0.713     | 0.807     | 0.045     | 2.958     | 9.620      |
> | MoGenTS   | 0.529     | 0.719     | 0.812     | **0.033** | 2.867     | 9.570      |
> | MotionGPT | 0.492     | 0.681     | 0.733     | 0.232     | 3.096     | 9.528      |
> | Ours      | **0.553** | **0.747** | **0.837** | 0.208     | **2.725** | 9.700      |
>
>
> **Beyond these numbers**, our choice of a continuous VAE latent is motivated by two representation-level advantages for generative motion modeling:
>
> - **Training stability and efficiency.** As shown in Tab. 5 (Appendix Sec. A), replacing our VAE with a VQVAE in exactly the same framework leads to slower convergence and worse motion generation/ caption quality, even with substantially longer training.
> - **Semantically smooth latent space.** Compared to quantized representations (VQVAE, RVQVAE, 2D quantization), the continuous VAE latent avoids quantization error and forms a smooth manifold, which naturally supports interpolation between motions, robust random sampling, and enables fine-grained via simple latent operations (e.g., interpolation, perturbation).
>
> In summary, while RVQVAE and 2D VQ are stronger reconstruction encoders, our experiments indicate that a continuous VAE latent offers a better trade-off for the generation and understanding tasks we study, yielding more stable training, better text–motion alignment and distributional metrics, and a smooth latent space that facilitates interpolation, sampling, and controllable motion generation in future work.

---

> ### Author Response · Authors · 2025-11-24
> **Reply to Reviewer Hcm9 (2/2)**
>
> ### W2. on more diverse datasets (Motion-X, KIT-ML).
>
> > W2. Experiments are only conducted on the HumanML3D datasets. Adding more diverse datasets, e.g., Motion-X and KIT-ML, will better illustrate the generalizability of the proposed framework.
>
> We agree that evaluating on more diverse datasets is important for assessing the generalizability of our framework. In addition to HumanML3D, we trained a VAE-based model and the corresponding T2M model (MotionGPT3†) on KIT-ML and report:
>
> **Table R3:** Motion generation on KIT-ML dataset. We also report the reconstructioin metrics for VAE used in MotionGPT3$\dagger$.
>
>
>   | Methods   | R@1↑    | R@2↑    | R@3↑   | FID↓  | MMDist↓   | Diversity↔ | MModality↑ |
>   | ---- | --- | ---- | --- | ---- | ---- | ----- | ----- |
>   | GT      | 0.424     | 0.649     | 0.779     | 0.031     | 2.788     | 11.08      | -      |
>   | VAE         | 0.4085 | 0.6273 | 0.7626 | 0.1548 | 2.855 | 10.9193 | -      |
>   | MotionGPT3$\dagger$ | **0.4549**| **0.6795**| **0.8027**| 0.2271| **2.7043** | *11.0255* | 0.9036   |
>   | MLD           | 0.390     | 0.609     | 0.734     | 0.404     | 3.204   | 1.080    | 2.192    |
>   | MotionGPT     | 0.366     | 0.558     | 0.680     | 0.510     | 3.527     | 10.350     | *2.328*    |
>   | MotionDiffuse    | 0.417   | 0.621   | 0.739   | 1.954   | 2.958   | **11.100** | 0.730    |
>   | ReMoDiffuse       | 0.427   | 0.641   | 0.765   | *0.155*   | 2.814   | 10.800     | 1.239    |
>   | MoMask        | 0.433   | 0.656   | 0.781   | 0.204     | 2.779     | -      | 1.131      |
>   | MoTe         | 0.419   | 0.627   | 0.741   | 0.256     | 3.216     | -      | **2.615**  |
>   | MoGenTS      | *0.445* | *0.671* | *0.797* | **0.143** | *2.711* | 10.918     | -     |
>
> These results show that our framework can be transferred to a different motion domain and maintains competitive text-to-motion performance, indicating the generalizability of our architecture.
>
>
> Regarding Motion-X, it is a very recent large-scale dataset and, to our knowledge, there is *not yet* a widely adopted official evaluation protocol or toolkit that would allow fair, directly comparable reporting across methods. To avoid introducing potentially incomparable metrics, we focus our current rebuttal on HumanML3D and KIT-ML, and leave a more extensive cross-dataset study on recent larger-scale benchmarks such as Motion-X or MotionMillion as an important direction for future work.

---

> > ### Author Response · Authors · 2025-11-26
> > **Updated results on KIT-ML**
> >
> > In our previous response, due to time constraints, we only shared our *initial* results on the KIT-ML dataset, which showed moderate performance due to an under-optimized VAE.
> > So far, we have fine-tuned the VAE on KIT-ML, which has led to significant improvements in subsequent generation tasks, surpassing state-of-the-art methods. For example, compared to MoGenTS, our model achieved **R@1 of 0.4549** (vs. 0.445) and **MMDist of 2.7043** (vs. 2.711).
> > The updated results, now shown in Table R3, demonstrate the effectiveness of these optimizations and further highlight the generalizability of our framework.

---

### Author Response · Authors · 2025-11-24
**Overall Clarification**

We thank the reviewers for the valuable feedback. We would like to clarify that our contributions lie not in any single component, but in the integrated combination of continuous motion representations, a decoupled bimodal design, and a three-stage alignment strategy.

Integrating these elements for motion data is non-trivial, and presents a substantial departure from prior text–image models or approaches based on discrete motion token. In response to the feedback, we have included additional quantitative comparisons(more datasets & more baselines), training analyses, inference latency results, and corresponding clarifications.

## **Motivation&Novelty**

We introduce an efficient bimodal motion–language model that enables both text-to-motion generation and motion-to-text understanding within a unified framework.

While individual components may draw inspiration from existing work, their synergistic integration for motion is novel, yielding clear empirical improvements. Our primary contributions include:

- **Continuous latent motion representation with a diffusion bridge.**

  Although VAE is a standard tool, adapting continuous motion latents to an autoregressive LLM is non-trivial.

  We leverage a lightweight diffusion head that bridges transformer hidden states and VAE latents, which significantly outperforms direct latent regression (Tab. 11(a) vs. (g), Appendix Sec. C.5) and enables both better quality and faster feed-forward inference in motion generation (Appendix Sec. A).

  **Table 11 (part): Ablation on motion supervision.**
  | ID  | Loss | R@1 ↑  | R@2 ↑  | R@3 ↑  | FID ↓  | MMDist ↓  | DIV →  | MModality ↑ |
  | ---- | ---- | --- | --- | ---- | ---- | ---- | --- | --- |
  | (a)  | Diff | **0.547** | **0.751** | **0.850** | **0.149** | **2.578** | **10.041** | **2.265**   |
  | (g)  | MSE  | 0.518     | 0.725     | 0.823     | 0.276     | 2.705     | 9.758      | 2.175       |

- **Dual-branch architecture to reduce cross-modal interference.**
  Unlike MotionGPT and MotionGPT-2, which use quantized motion codes within a single-stream backbone, and BAGEL, which adopts task-wise dual-branch architecture (i.e., one branch for understanding and one for generation, with seperate experts and adaptors), MotionGPT3 adopts a *modality-wise* dual-stream design where the motion and text branches have separate parameters and interact only via cross-modal attention.

  This design mitigates gradient interference across modalities, improves convergence stability (Fig.3, Tab. 3), and allows task-specific configurations for motion branch, such as attention mechanisms and model size (Appendix Sec. C.3 and C.4).


  **Table 3:  Ablations for representation choice and architecture design. Unified denotes a single-stream backbone and Bimodal denotes a dual-stream backbone.**

  (a) Comparison on Text-to-Motion.

  | Setting  | R@1  | R@3  | MMDist    | FID    |
  | ------ | ---- | ----- | -- | --- |
  | Unified+VQ  | 0.237     | 0.435     | 5.684     | 0.403     |
  | Bimodal+VQ  | 0.300     | 0.532     | 4.937     | 0.454     |
  | Unified+VAE | 0.501     | 0.792     | 2.841     | 0.489     |
  | Bimodal+VAE | **0.533** | **0.826** | **2.797** | **0.239** |

  (b) Comparison on Motion-to-Text

  | Settings    | R@1    | R@3    | MMDist | BertScore  |
  | ----- | ----- | ---- | ------ | ----- |
  | Unified+VQ  | -         | -         | -      | -          |
  | Bimodal+VQ  | 0.379     | 0.702     | 3.545  | 18.085     |
  | Unified+VAE | 0.234     | 0.426     | 5.976  | 16.197     |
  | Bimodal+VAE | **0.553** | **0.853** | 2.524  | **35.850** |

- **Three-stage training strategy for stable multimodal alignment.**

  We observe that multi-task training from scratch severely deteriorates both T2M and M2T performance (Two-Stage training in Tab. 4).

  Our three-stage schedule, with (i) generation pretraining, (ii) cross-modal alignment, and (iii) joint finetuning, first grounds the motion branch and then aligns it with the text representation, leading to more stable optimization (Sec. 3.4; Fig. 2).



  **Table 13 (part): Ablation on three-stage training scheme.**

  | Type                | Stage 1 | Stage 2 | Stage 3 | T2M R@3↑  | T2M FID↓  | T2M MMDist↓ | M2T R@1↑  | M2T Bleu@4↑ | M2T BertScore↑ |
  | ------------------- | ------- | ------- | ------- | --------- | --------- | ----------- | --------- | ----------- | -------------- |
  | Three Stages        | ✓       | ✓       | ✓       | **0.837** | **0.208** | **2.725**   | **0.573** | **19.412**  | 35.231         |
  | Two Stages          | ✗       | ✓       | ✓       | 0.772     | 0.325     | 3.108       | **0.573** | 18.277      | **35.546**     |
  | Trained Text Branch | ✓       | ✓       | -       | 0.801     | 0.243     | 2.942       | 0.505     | 14.119      | 33.385         |


Together, these components form a robust and scalable bimodal framework, capable of high-quality motion generation and understanding.

---

### Author Response · Authors · 2025-11-30
**Summary of Rebuttal**

Dear Area Chairs, Senior Area Chairs and Program Chairs,

Thank you very much for taking the time to review our submission.

We sincerely thank the reviewers for their positive response and recognition of our core contributions, including the motivation to bridge the gap between motion and language, and the targeted design of  continuous latent representations along with the dual-branch architecture (Reviewers Hcm9, 9qBu, cYVn).
We are grateful for the acknowledgment of the significant quantitative/qualitative performance gains achieved by our method (Reviewers Hcm9, VYUy, cYVn, 9qBu).
Additionally, we appreciate the reviewers' favorable comments about the clarity of the paper (Reviewers Hcm9, 9qBu), the detailed visualizations, the thorough ablation studies (Reviewers VYUy, cYVn, Hcm9), and the reproducibility (Reviewer cYVn).

In the rebuttal, we have carefully addressed all concerns of reviewers point-by-point with (i) additional experiments on KIT-ML datasets, more language backbones, and inference latency,  (ii) further analysis on continuous VAE latent and discrete token, (iii)  implementation details of training and inference procedure, and (iv) discussion on recent works with autoregressive continuous token or bimodal backbone.

By November 26, we have received positive follow-up feedback from Reviewer 9qBu, who explicitly acknowledged the adequacy of our rebuttal. The other reviewers did not respond, so we were unable to further engage in clarification with them.

Below we summarize the main clarifications:

1. **Motivation & novelty** (Overall Clarification)

    Our contribution lies in the *synergistic integration* of (i) continuous latent modeling, (ii) modality-separated dual-stream, and (iii) a stable three-stage alignment strategy.
    Together, these designs enabling a robust motion–language model for bidirectional motion–language modeling.
    To the best of our knowledge, such a unified LLM-style framework operating on continuous motion tokens has not been explored in prior work.

2. **Generalization across datasets and backbones** (Reviewers Hcm9, 9qBu)

    *Datasets.* We provided T2M comparisons on KIT-ML in *Table R3* (surpass SOTA), which demonstrates our generalizability beyond HumanML3D.

    *Backbones.* We evaluate our model with GPT-2, Flan-T5-base, and Qwen2.5-0.5B-Instruct, results summarized in *Table R7*. Our dual-stream design consistently improves T2M metrics, indicating that our framework can scale to more recent LLMs.

3. **Continuous VAE latent vs. VQ-based encoders** (Reviewer Hcm9)

    While recent RVQVAE/2D-VQ methods achieve strong reconstruction, our continuous latent offers smoother semantic, better training stability, and superior T2M/M2T performance (*Tables R1&R2*).

4. **Dual-stream architecture** (Reviewers VYUy, cYVn)

    Different from single-stream MotionGPT/MotionGPT-2 or task-specific branches in BAGEL, our modality-specific dual-stream framework effectively reduces cross-modal interference, with improved metrics of R-Precision and MMDist, as shown in *Table 3&R4*.

5. **Recent baselines** (Reviewer cYVn)

    We compared MotionGPT3 with strong recent models such as MotionGPT-2, LaMP and MG-MotionLLM in *Table R9-R11.* MotionGPT3 remains highly competitive, it consistently outperforms all compared methods on MMDist, and all T2M metrics under the TMR evaluator, reflecting stronger text–motion alignment.

6. **Inference latency** (Reviewers 9qBu, cYVn)

    Detailed analysis in *Table R12* shows that our design with continuous VAE and lightweight diffusion head improves T2M inference efficiency.

7. **Training details for three-stage alignment and inference settings** (Reviewer VYUy)

    We provide a clear explanation of the three training stages as well as the inference process. Ablations in *Table 13* indicate that the proposed three-stage scheme is essential for stable optimization and outperforms two-stage or fully joint training.

We hope these clarifications help you quickly grasp the core discussions with the reviewers and the contribution of our work. Thank you again for your time and consideration.

---

### Meta-Review · Area_Chair_F1h9 · 2026-01-03

**Summary:**

This submission proposes MotionGPT3, a bimodal motion-language framework intended to unify motion understanding and motion generation, via (i) a continuous VAE motion latent coupled with a lightweight diffusion head, (ii) a dual-strem Transformer with shared attention for cross-modal exchange, and (iii) a three-stage generate-then-align training schedule for better optimization and convergence. It accept borderline ratings. Reviewers generally found the motivation clear and the empirical results strong on the main benchmarks, with two reviewer rating it 6, and two rating it 4.

The core concerns that kept the paper borderline are:
* **Evidence for the continuous VAE vs. recent dicrete/quantized encoders** was initially missing (e.g., MoMask, MoGenTS, MotionGPT2), and reviewers questioned whether the claimed benefits were substantiated.
* **Generalization beyond HumanML3D** was quoted, with requests for broder benchmarks (KIT-ML, Motion-X) and concern that performance might not transfer robustly.
* **Efficiency/latency reporting** was requested, and some reviewers also noted incomplete discussion&analysis against recent baselins/LMs.
* Two reviewers also raised the concern that the approach was **incremental/overly complex** relative to MotionGPT/MotionGPT2 and asked for clearer exposition of the pipeline and reasoning.

**Reviewer Concerns:**

### *Continuous VAE vs. discrete/quantized motion encoders (MoMask/MoGenTS)* (Reviewer hcm9)
* **Partially Addressed**
* Authors provided reconstruction comparisons (showing quantization-based models reconstruct better) and argued the continuous latent offers a better trade-off for their generation/understanding goals, with stronger alignment metrics on the main T2M benchmark; they also acknowledge reconstruction is currently weaker and attribute this to an under-optimized VAE.
* Remaining gap: the reconstruction performance is notably underperforming existing VQ methods. Though the final generation alignment gets improved, the quality metric (FID) is actually much worse than existing SToA (e.g., MoMask, MoGenTS) which might be attributed to the inferior representation.

### *Generalization beyond HumanML3D (KIT-ML / Motion-X)* (Reviewer hcm9)
* **Partially addressed**
* Authors added KIT-ML results and claimed better performance than MoGenTS on retrieval metrics.

### **Novelty / conceptual contribution (incremental; similarity to prior bimodal frameworks)** (Reviewer VYUy, cYVn)
*  **Partially addressed**
*  The authors elaborated further that their novelty lies in the continuous diffusion representation in a bi-modal dual-stream Transformer that unifies the task of M2T, T2M and M2M. They also clarified that the key difference bettween their work and BAGEL is that dual-stream are *modality-wise* while the branches in BAGEL are *task-specific*, as well as a few other adaptations for motion tasks. Although “novelty vs. complexity” judgment remains somewhat subjective, the positions and contributions of current work seem quite clear, and recognized by the other two reviewers (Hcm9, 9qBu)

### *Efficiency / inference latency / FLOPs (and per-component overhead)* (Reviewer 9qBu, cYVn)
* **Addressed**
* Authors respond with a latency breakdown across architectures and tasks (Table R12).

### *Baselines outdated / LM choice (e.g., more recent models than MotionGPT-2; recent motion-language baselines)* (Reviewer 9qBu, cYVn)
* **Addressed**
* The auhors added comparisons with MotionGPT2, MG-MotionLLM as well as more LM backbones.

### *Metric interpretation: lower MModality* (Reviewer 9qBu)
* **Addressed**
* Authors clarified that (i) overall Diversity remains comparable, and (ii) MModality can decrease as text-conditioning becomes more semantically consistent (tighter conditional clusters), so it does not necessarily imply mode collapse.

### *Clarity of method/training pipeline description (esp. Fig. 2 / stage design)* (Reviewer UYUy)
* *addressed*

**Reviewer Scores:**

Reviewer 91Bu explicitly state most concerns were addressed and they are inclined to maintain original positive score (6).

Reviewer Hcm9 may likely remain the original rating (6), as their concerns (broader comparisons with more representation and datasets) were addressed in the rebuttal.

Reviewer VYUy acknowledged that scores can be increased if the authors address their questions.  In the rebuttal, the authors expand the training details of three stages, and clarify the position of current work compared to MotionGPT/MotionGPT2, and stressed their novelty beyond merely using continuous diffusion representation, which are recognized in the strengthes in two other reviewers' comments. It's likely that the reviewer would increase the score to 6.

Reviewer cYVn primarily holds three concerns: 1) similarity to BAGEL, 2) outdated baselines, 3) lack of inference latency evaluation. In the rebuttal, the authors distinguish their works to BAGEL, and provided extra evaluation of latency and comparing to more baselines (MotionGPT2, MG-MotionLLM). Given most of the concerns are answered well, I would expect the reviewer raising their score to borderline/borderline accept.

Therefore, I recommend acceptance of this work.

---

### Decision · Program_Chairs · 2026-01-26

Accept (Poster)